# CLAP4CLIP: Continual Learning with Probabilistic Finetuning for Vision-Language Models

**Saurav Jha**[1], **Dong Gong**[1]*, **Lina Yao**[1,2]

[1]University of New South Wales (UNSW Sydney), [2]CSIRO's Data61

{saurav.jha, dong.gong}@unsw.edu.au; lina.yao@data61.csiro.au

## Abstract

Continual learning (CL) aims to help deep neural networks learn new knowledge while retaining what has been learned. Owing to their powerful generalizability, pre-trained vision-language models such as Contrastive Language-Image Pre-training (CLIP) [1] have lately gained traction as practical CL candidates. However, the domain mismatch between the pre-training and the downstream CL tasks often calls for finetuning of the CLIP on the latter. Most existing finetuning methods exhibit deterministic nature. This makes them overlook the many possible interactions across the input modalities and deems them unsafe for high-risk tasks requiring reliable uncertainty estimation. To address these, our work proposes **C**ontinual **LeA**rning with **P**robabilistic finetuning (CLAP) - a probabilistic modeling framework over visual-guided text features per task, thus providing more calibrated CL finetuning. Unlike recent data-hungry anti-forgetting CL techniques, CLAP alleviates forgetting by exploiting the rich pre-trained knowledge of CLIP for weight initialization and distribution regularization of task-specific parameters. Cooperating with the diverse range of existing prompting methods, CLAP can surpass the predominant deterministic finetuning approaches for CL with CLIP. We conclude with out-of-the-box applications of superior uncertainty estimation abilities of CLAP including novel data detection and exemplar selection within the existing CL setups. Our code is available at `https://github.com/srvCodes/clap4clip`.

## 1 Introduction

Learning in the real world involves dealing with the ever-changing distributions of task streams and their data [2, 3, 4]. Given the constraints on resources and privacy, there is also no guarantee for re-training a network on all previously seen data [5]. Continual learning (CL) aims to learn from such data/task stream without *catastrophic forgetting* [6, 2] of past data/tasks. A challenging CL setup is the class-incremental learning setting, where new classes emerge with new tasks, and at test time, a model must infer from all seen classes without known task IDs [7, 8].

Recent years have seen pre-trained multi-modal foundation models excel on several domains [1, 9, 10]. One such example for the vision-language (VL) domain is the CLIP [1] model that comes with strong zero-shot generalizability acquired by learning to match large-scale image-text pairs in a contrastive manner [11]. However, to adapt well to downstream tasks, CLIP must be finetuned on the task-specific data [12, 13]. Considering both the need for continually finetuning pre-trained models on streaming tasks and their perks over training from scratch [14], our work studies CL with CLIP.

An issue with the existing deterministic approaches to finetuning [13, 12] is that these overlook the *uncertainties* arising from many possible interactions between the visual and textual cues of downstream tasks. For instance, on the textual side, while a good generic hand-crafted prompt for images is "`A photo of a {class}`", there can be instances where further tailored prompts help

---

*Corresponding author

38th Conference on Neural Information Processing Systems (NeurIPS 2024).

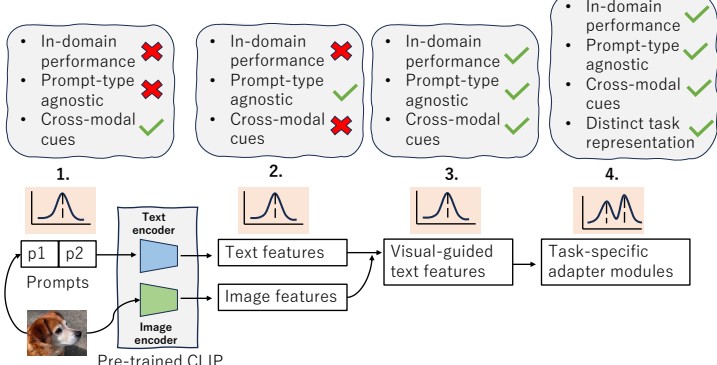

Figure 1: **Concept diagram for probabilistic finetuning of pre-trained CLIP in a CL setup:** We identify four such suitable design choices for probabilistic modelling. Choice #1 [17] performs variational modelling by imposing prior on the prompt space which makes it prompt-type dependent while also interfering with the in-domain knowledge learning capability of the prompts – a criterion crucial in the deployment of CL models. Choice #2 (see Sec. 3.2.1) instead imposes a prior on the outputs of the text encoder. While this makes it prompt-type agnostic (as the text features can now be derived from arbitrary prompt types), not taking the visual information into account nevertheless leads to the loss of information about cross-modal interactions between the visual and textual cues – a property essential for preventing cross-modal deviation of finetuned features in CL (see Sec. 3.2.2). Choice #3 (Ours) leverages the best of both worlds by modelling the distribution of visual-guided text features. To further refine the learned distributions of CL tasks, we finally introduce lightweight task-specific adapter modules in choice #4 that make the cross-task centroids more distinct while preserving the aforesaid properties (see Sec. 3.2.3).

improve the image-to-text coherence. Similarly, on the visual side, images from the same class can have diverse range of backgrounds, poses, orientations, etc. Overlooking the uncertainties in image-text matching can thus cause overfitting on downstream tasks and forgetting of the generalizable knowledge [15]. For CL, where we seek to adapt CLIP on a stream of tasks, this can further lead to the cross-modal features increasingly deviating from each other to the point of catastrophic forgetting (see Fig. 3b). While existing methods [16, 17] model such uncertainties through probabilistic finetuning, these remain subpar at CL given: (a) their inaptness to leverage existing prompt-based approaches [16], (b) their excessive trading of in-domain performance for generalization [17].

Lastly, like other autonomous real-world agents, CL models deployed in mission-critical settings (healthcare, transport, etc.) can benefit from uncertainty awareness by calibrating their predictions to reliably assess their confidences [18, 8]. Hence, to enhance the usage of the pre-trained CLIP for real-world CL tasks, we design a finetuning approach with the following three properties (see Fig. 1): A) **probabilistic modeling** of cross-modal task cues for better generalization; B) **compatibility** with prompt-based finetuning methods [14, 12, 19, 20] to exploit their finegrained in-domain task knowledge; C) leveraging the **rich pre-trained knowledge** of CLIP to further counter forgetting.

To this end, we design a principled Variational Inference (VI) framework that learns the functional space of task-specific posterior distributions based on text features that are aligned with their visual counterpart (property #**A**). To refine these variational task posteriors, we draw our motivation from Bayesian mixture-of-experts ensembles [21, 22], and employ lightweight task-specific adapter modules that model task-specific distributions. Our final prediction is thus a mixture of logits derived from individual task adapter modules. To further counter the forgetting in these modules, we diverge from the recent trend of internet data-hungry CL techniques [23, 24]. Instead, we exploit the readily available pre-trained language knowledge of CLIP for weight initialization and task distribution regularization (property #**C**). Finally, by modelling the distribution of the text feature space, our probabilistic finetuning method boasts a rich modular nature as the features can be derived from arbitrary prompt types (property #**B**). In particular, we show that our framework can inherit the in-domain knowledge of hand-crafted [1], uni-modal [12], multi-modal [20], or instance-conditioned [19] prompts. We backronymize our finetuning approach as **CLAP** – **C**ontinual **LeA**rning with **P**robabilistic finetuning – for the pre-trained CLIP model. Our experiments across

several settings show that CLAP4CLIP enhances *prompt-based* finetuning for CLIP, and surpasses the predominant deterministic finetuning methods in terms of in-domain performance, output calibration, and generalization to unseen CL tasks, all while sharing a similar resource overhead. We study some out-of-the-box perks of CLAP's probabilistic nature by leveraging its uncertainty estimation capabilities on a proposed *post-hoc* novel data detection setup and on exemplar selection for CL.

## 2   Related work

**Continual Learning (CL).** The existing CL literature is predominated by three categories of methods: (a) Regularization-based methods [6, 25, 26] alleviate forgetting by punishing changes to the parameters that are important to previous tasks; (b) Architecture-based approaches learn parameters that are specialized for individual tasks either by network expansion [27] or by sub-network composition [28, 29]; (c) Rehearsal-based approaches [30, 5] rely on storing a fraction of the past task experiences in a memory to train with the current task. Each category has its own flaw – methods in (a) struggle to discriminate inter-task classes [31]; those in (b) often require task oracle during inference; those in (c) are sensitive to the memory sizes besides being prone to overfitting on the memory samples [32]. Hence, practical CL calls for combining these. Our work leverages (a) via function-space regularization (Sec. 3.4), (b) via task-specific modules (Sec. 3.2.3), and (c) via herding-based replay (see App. A.1.1).

**Vision-Language Models (VLMs) finetuning.** The powerful generalizability of pre-trained VLMs [9, 33] like the CLIP [1] has enabled their zero-shot applications to a range of downstream tasks, including CL [14]. In practice, their performance on downstream out-of-domain data remains rather weak [34, 35]. For such cases, finetuning on task-specific data is a natural choice. Instead of performing extensive *full finetuning* on all parameters, some *parameter-efficient finetuning* (PEFT) methods learn a lightweight feature adapter module for textual and/or visual paths [13, 36]. Another line of PEFT methods learns *soft prompts* which are a few continuous tokens serving as inputs to the frozen visual and/or textual encoder(s) to capture task-specific information [12, 37, 20]. Existing works on CL with pre-trained CLIP have leveraged either [19, 38] or both [39] of these methods. However, such finetuning methods remain deterministic in nature. This imposes an explicit constraint on the modeling of the possible ways in which the visual and the textual semantics interact.

To address the aforesaid flaw, one could turn to adapt the existing probabilistic finetuning approaches to capture the cross-modal interactions in CL tasks. For instance, [16] learn the distribution of hand-crafted prompts while [17] propose variational prompt tuning (VPT) *conditioned* on the input images. Yet these methods are limited in their efficacy. [16] is incompatible with conditional prompt learning [37], which is an extensively studied PEFT field. VPT [17] trades in-domain performance excessively in favor of generalizability – a trait detrimental in the deployment of CL models. Our work aims to bridge these gaps for probabilistic finetuning all while adapting it for CL.

## 3   Methodology

### 3.1   Preliminaries

**Continual Learning (CL).** Class-incremental CL [7] aims to learn from a sequence of $T$ tasks $[(C^1, D^1), (C^2, D^2), ..., (C^T, D^T)]$. Each task $t \in [1, T]$ has its training data $D^t = \{(\mathbf{x}_1, y_1), (\mathbf{x}_2, y_2), ..., (\mathbf{x}_{k^t}, y_{k^t})\}$, where $\mathbf{x}$ and $y$ are the input images and labels, respectively from the set of classes $C^t = \{c_1^t, c_2^t, ..., c_{n^t}^t\}$. Following [40, 41], we assume any two task-specific sets of classes to be disjoint: $C^i \cap C^j = \emptyset$. A neural network with parameters $\phi$ is then trained on task $t$ using $D^t$ to minimize the cross-entropy loss over $C^t$. At test time, the model is evaluated on all seen classes $C = \bigcup_{i=1}^{t} C^i$, where the past task predictions are prone to forgetting. As a solution, rehearsal-based methods [42, 43] replay past task samples from a memory $\mathcal{M}$ during training. Following [30], we use herding [44] to maintain $\mathcal{M}$ (see App. A.1.1 for details).

**CLIP with prompt.** CLIP comprises an image encoder $f(\mathbf{x})$ acting on an image $\mathbf{x} \in \mathbb{R}^{3 \times H \times W}$ of height $H$ and width $W$, and a text encoder $g(\mathbf{t}(\mathbf{p}))$ acting on a word embedding vector $\mathbf{t} \in \mathbb{R}^{(L \times e)}$ derived from a text prompt $\mathbf{p} \in \mathbb{R}^{((L-1) \times e)}$. Here, $L$ is the text length, and $e$ is the text embedding dimension. The encoders's outputs are used in the prediction of the class $y_i$ as:

$$p(y_i|\mathbf{x}) = \frac{\exp\left(\langle f(\mathbf{x})^T, g(\mathbf{t}_i)\rangle/\tau\right)}{\sum_{c=1}^{|C^t|} \exp\left(\langle f(\mathbf{x})^T, g(\mathbf{t}_c)\rangle/\tau\right)}, \tag{1}$$

where $\tau$ is a learnable temperature parameter, $\langle \cdot, \cdot \rangle$ is the cosine similarity, and the $c$-th class text feature $\mathbf{t}_c = [\mathbf{p}, \mathbf{e}_c]$ is the result of adding a class-specific word embedding $\mathbf{e}_c$ to the prompt $\mathbf{p}$. The

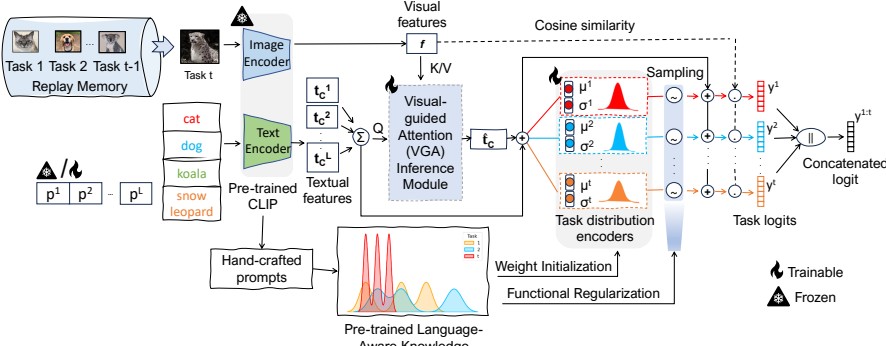

Figure 2: **CLAP4CLIP overview:** the visual-guided attention (VGA) inference module uses the text features as query (Q), and the visual features as keys (K) and values (V) to produce visual-guided text features. The task-specific text features are fed to their respective task distribution encoders $(\mu^t, \sigma^t)$. The task distribution samples are then fused with the original task features prior to deriving the task logits $y^t$. All task logits are concatenated to produce the final prediction $y^{1:t}$.

features $g(\mathbf{t}_c)$ for all classes are used as the weights of a linear classifier. In CL, $g(\mathbf{t}_c)$ would thus encode the features for classes $c \in C$ seen until task $t$. Eq. (1) forms a contrastive training criterion for the text and visual modalities, whose rich representation allows pre-trained CLIP to be used for zero-shot classification through *hard* prompt templates, *i.e.,* $\mathbf{p}^c =$ "A photo of a {c$^{\text{th}}$ class}".

**CLIP finetuning with learnable soft prompts.** To improve the CLIP performance on a downstream task $t$, *soft* prompts use a set of learnable vector tokens $\mathbf{p} = \{\mathbf{p}^1, \mathbf{p}^2, ..., \mathbf{p}^L\}$. CoOp [12] shares $\mathbf{p}$ with all classes of a task. MaPLe [20] learns multi-modal prompts by employing two such token sets $\mathbf{p}_f$ and $\mathbf{p}_g$ until the $J$-th layers of the vision and the text encoders of CLIP, respectively. AttriCLIP [19] selects a subset of prompts conditioned on the input: $\{\{\mathbf{p}^j\}_{1 \le j \le L} | \mathbf{x}_k\}$. Learning $\mathbf{p}$ (with frozen CLIP weights) thusly helps encode task/modality/instance-conditioned context for a given task.

**CLIP finetuning with adapters.** Adapter-based methods like CLIP-Adapter [13] learn lightweight modules over text and/or visual features of the frozen CLIP model. With a text adapter $A_t$, the updated text features from Eq. (1) can be rewritten (with a slight abuse of notation) as:

$$g(\mathbf{t}_i) = \alpha A_t(g(\mathbf{t}_i)) + \beta g(\mathbf{t}_i), \tag{2}$$

where $\alpha$ and $\beta$ control the strength of the residual connection between the adapted and the pretrained models' features, e.g., $\beta = 1 - \alpha$ in [13].

### 3.2 CL with probabilistic finetuning for CLIP

**Overview.** We develop our CLIP-based probabilistic finetuning model using a Bayesian VI framework (see Fig. 2). Sec. 3.2.1 starts by making the case that, unlike previous VI-based finetuning approaches, the feature embedding space of the text encoder output is a superior choice for defining our functional space priors on. While linear adapter layers are often employed for obtaining the mapping from such a pre-defined feature-space prior to the function outputs [45, 17], we show in Sec. 3.2.2 that CL finetuning with generic adapter leads to the issue of cross-modal deviation. In Sec. 3.2.3, we then propose refining our variational distribution on function space using an ensemble of task-specific adapters that are built on top of cross-modal aligned text features.

#### 3.2.1 Variational inference with function space prior on text features

We are interested in modeling the stochastic processes that generate the labels $y$ for the inputs $\mathbf{x}$ of a CL task $t$. To this end, we assume a prior distribution $p_\chi$ over the text feature of the $c$-th class: $p_\chi(\mathbf{t}_c(\mathbf{p}))$. We can then draw $M$ number of latent variables $z = \{z_m \sim p_\chi\}_{m=1}^M$ to represent the $c$-th class text feature $\mathbf{t}_c(\mathbf{p})$ as a linear combination of the text encoder feature $g(\mathbf{t}_c(\mathbf{p}))$ and $z$:

$$\mathbf{t}_c(\mathbf{p}) = \{g(\mathbf{t}_c(\mathbf{p})) + z_m\}_{m=1}^M, \quad \text{s.t. } z_m \sim p_\chi, \tag{3a}$$

$$p(y_i|\mathbf{x}) = \int_\chi \frac{\exp\left(\langle f(\mathbf{x})^T, \mathbf{t}_c(\mathbf{p})\rangle\right)}{\sum_{c=1}^{|C^t|} \exp\left(\langle f(\mathbf{x})^T, \mathbf{t}_c(\mathbf{p})\rangle\right)} p(\mathbf{t}_c(\mathbf{p})) d\chi, \tag{3b}$$

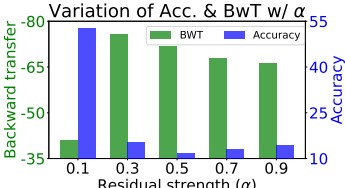
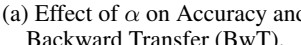
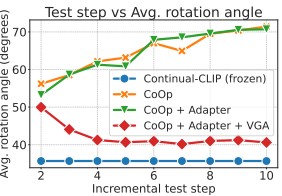

(a) Effect of $\alpha$ on Accuracy and
Backward Transfer (BwT).

(b) Avg. rotation angle [47] per incremental step
for image and text features.

Figure 3: **Need for Visual-guided Attention (VGA) inference module.** Fig. 3a: A simple adapter is inadequate at preventing catastrophic forgetting in CL – marked by high BwT scores; Fig. 3b: VGA module encourages cross-modal alignment between the learned text features and the pre-trained visual features – marked by a decrease in average angle $\texttt{arccos}\langle t, 1\rangle$ between them – where otherwise the former deviates further with incremental training steps.

where Eq. (3b) replaces $g(\mathbf{t}_i)$ in Eq. (1) by $\mathbf{t}_c(\mathbf{p})$. Eq. (3b) thus results into $M$ number of predictions whose distribution gives us the model's epistemic uncertainty about the correct prediction. To deal with the intractability of the marginal likelihood, we optimize for the evidence lower bound (ELBO) using a variational posterior $q_\phi$ that approximates the prior $p_\chi$ based on the KL-divergence loss $\mathbb{D}_{KL}$:

$$\log p(y|\mathbf{x}) \geq \mathbb{E}_{q_\phi(z|\mathbf{t}_c)}[\log p(y|\mathbf{x}, z)] - \mathbb{D}_{KL}\big(q_\phi(z|\mathbf{t}_c)\|p_\chi\big). \tag{4}$$

By assuming $p_\chi$ to be the (static) standard Gaussian $\mathcal{N}(\mathbf{0}, \mathbf{I})$ and $q_\phi$ to be the (learnable) Gaussian $\mathcal{N}(\mu(\mathbf{t}_c), \sigma(\mathbf{t}_c))$, whose mean $\mu$ and standard deviation $\sigma$ are parameterized by linear adapter layers, we can ensure that the random variable $z$ remains differentiable by reparameterization trick [45]. Accordingly, we refer to the parameters $[\mu; \sigma]$ together as *probabilistic adapter* from here onward.

Imposing the prior over the text feature offers us *further advantages* over that in the prompt embedding space as done by VPT [17] (also see App. Fig. 1 for an illustration). First, Eq. (3b) is prompt-type agnostic as the text features $g(\mathbf{t})$ could be derived from any existing soft [37, 19, 20] or hard [14, 13] prompts.[2] Second, by leaving the prompt embedding space intact and injecting stochasticity into the feature space, we can better learn the task-specific knowledge known to be encoded by the prompt embeddings [46]. This can help bypass the loss of in-domain performance. In CL, the latter property is crucial for our model to perform well on all previously seen tasks. Third, as the latent variable $z$ is now directly used to infer the logits, it naturally favors generalization by influencing the predictions. On the contrary, the effect of the prompt space prior on the predictions is indirect as it is mediated by the representations of the entire text encoder layers. This can make the influence of the prior harder to control and can hinder the aforesaid interpretability in the model's predictions.

**Efficient continual finetuning with a probabilistic adapter.** Finetuning adapters for CL is not straightforward. Namely, we have the overhead of searching for task-specific residual ratio $\alpha$ (see Eq. (2)) which is sensitive to the training setup including dataset and prompt-type [13, 36]. This has particularly worse implications for a probabilistic adapter like ours, where a larger $\alpha$ can inject enough noise to corrupt the pre-trained representations to the point of catastrophic forgetting (see Fig. 3a). For efficient learning of our adapter, we thus seek to retain *no additional overhead* of hyperparameter search for the residual ratio. Subsequently, we use $\alpha = \beta = 1$ through our work.

### 3.2.2 Cross-modal feature deviation in continual finetuning of CLIP

To perform CL with variational modelling in the text feature space, we first take a step back to investigate how CL in general affects the *cross-modal deviation* [47] between the learned text and the frozen visual features of finetuning methods. To this end, we consider two basic CL models: the CoOp [12] and the CoOp with a CLIP-Adapter [13]. Then, for the base task ($t = 1$) test samples of CIFAR100, we compute the average of the Rotation Angle Matrix (RAM) [47] using the CL models' frozen visual $f(\mathbf{x})$ and learnable textual $g(\mathbf{t}_c(\mathbf{p}))$ features at each incremental test step. Fig. 3b shows the deviation of the learned textual features from their (frozen) visual counterparts for the CoOp. This implies that the cross-modal retrieval performance of CLIP finetuned with learnable prompts deteriorates with incremental training. Moreover, as a generic adapter (CoOp + Adapter) does not remedy the cross-modal deviation, this sets a direct hindrance in employing our probabilistic adapter to learn the variational distribution $q_\phi$.

---

[2]From a practitioner's perspective, this enriches the modularity by taking away the overhead of engineering the priors specific to the prompt-type that could, for instance, be spanning multiple layers and/or modalities.

**Variational modeling on visual-guided text features.** For variational modeling of text features $\mathbf{t}_c(\mathbf{p})$ that remain aligned with the visual features during continual finetuning, we propose enriching the text features with the visual context through an explicit attention mechanism (see App. A.3 for further justification on this design choice). To this end, we treat the text features as queries $Q$ and the visual features as keys $K$ and values $V$, and adopt a standard transformer-styled decoder block [48] as a task-shared Visual-guided attention (VGA) module. The VGA module performs text-to-text self-attention followed by text-to-visual cross-attention. To eliminate the influence of the text features from multiple CL tasks, a *naive* strategy is to perform specialized VGA forward passes with task-specific queries [27]. We seek to replace several such costly VGA passes with a single pass. To do so, we exploit the global nature of our visual context and mask out (set to $-\infty$) all inter-task connections in the queries using a target mask. This ensures that only the task-specific text features undergo self-attention while the entire query still attends to the visual context:

$$\{\hat{\mathbf{t}}_c^k\}_{k=1}^t = \text{VGA}\big(Q = \{\mathbf{t}_c^k\}_{k=1}^t, K = V = f(\mathbf{x})\big), \tag{5a}$$

$$\tilde{\mathbf{t}}_c^t = \hat{\mathbf{t}}_c^t + g(\mathbf{t}_c^t(\mathbf{p})), \tag{5b}$$

where $\hat{\mathbf{t}}_c^k$ is the task-specific visual-guided text feature which is fused with the residual task-specific text encoder feature $g(\mathbf{t}_c^t)$ to derive the task embedding $\tilde{\mathbf{t}}_c^t$. We note that in a non-CL setup, [49] employ the VGA module using the per-pixel spatial features (obtained before global average-pooling) instead of the globally pooled visual features of the ViT [50]. Our choice for the latter favors the efficiency of our framework for large CL datasets where attending to per-pixel spatial features can incur much higher latency (see App. A.4 for a comparison).

### 3.2.3 Task-specific probabilistic adapters as ensembles for posterior approximation

By exploring diverse modes in function space, ensembles of neural networks can better approximate the variational posterior [51, 22]. Motivated by this, we replace our task-shared adapter $q_\phi$ with task-specific adapters $\{q_\phi^i\}_{i=1}^t$ that parameterize the $t$-th task-specific posterior $\mathcal{N}(\mu^t, \sigma^t)$ over the task embeddings $\tilde{\mathbf{t}}_c^t$:

$$\{z_m^t\}_{m=1}^M \sim q_\phi^t(z|\tilde{\mathbf{t}}_c^t) = \mathcal{N}\big(\mu^t(\tilde{\mathbf{t}}_c^t), \sigma^t(\tilde{\mathbf{t}}_c^t)\big), \tag{6}$$

where $z_m^t$ are the task-specific MC samples. Task-specific adapters thus serve as mixture-of-experts ensemble where each expert is trained on task-specific embeddings $\tilde{\mathbf{t}}_c^t$ and the logits computed using each expert is combined to derive the final pre-diction $\hat{y}^{1:t}$ (see Algo 1). The experts learn posteriors that are more discriminative across tasks. This is depicted in Fig. 4 using the cosine distance between the embeddings of the class-specific samples drawn from the posteriors. With task-specific adapters (right), the cross-task class centroids are more separable.

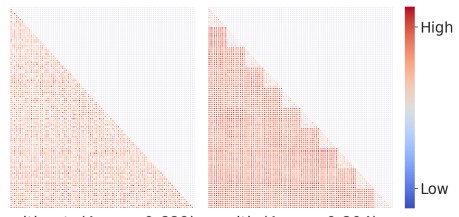

Effect of task-specific encoders on inter-class centroid distances

without (Avg. = 0.689)   with (Avg. = 0.804)

To prevent interference from current task training data, we freeze the past task encoders during each incremental training step ($t > 1$). Moreover, to reduce the forgetting in past task adapters, we follow other parameter-isolation techniques [27, 42, 52] to finetune on a class-balanced dataset of new data and rehearsal data $\mathcal{M}$ at the end of each incremental training step ($t > 1$). We refer to this as memory consolidation training (see App. A.2). We also provide an overview of a test-time forward pass of our framework in App. A.5.

Figure 4: **Need for task-specific probabilistic adapters:** Cosine distance between the centroids of class-specific latent variables produced without (**left**) and with (**right**) task-specific adapters on CIFAR100 (10 tasks, 10 classes per task).

**Algorithm overview.** App. algo. 1 outlines the pseudo-code of a forward pass of CLAP at $t-$th task test step. Here, a test image is to be classified into one of the classes $\{1, ..., |C^t|\}$. Our method executes the computationally heavy VGA layers only once. The task-specific VGA outputs are passed to their respective adapters. By limiting the quadratic complexity of the VGA pass, our method induces minimal time overhead. By only expanding the linear adapters per task, our memory overhead is negligible compared to the large backbone of the pre-trained CLIP model (ablation Fig. 5).

### 3.3 Alleviating forgetting with pre-trained language-aware CLIP knowledge

Like other finetuning methods, our probabilistic adapters are likely to trade the generalizability of text features for downstream task performances [12, 37]. On the contrary, the pre-trained CLIP text

encoder with hand-crafted prompts [53] has strong generalizability because of its rich pre-trained language information. We propose to leverage this pre-trained language knowledge to help guide the incremental finetuning in CLAP. In the following, we assume $\{\mathbf{t}_y^{h,l} \in \mathbb{R}^d\}_{l=1}^L$ to be the features corresponding to the $L$ hand-crafted textual prompts for the class $y \in C^t$.

### 3.3.1 Past-task distribution regularization for mitigating forgetting

The functional spaces of the past task distributions are prone to forgetting in CL. Though replay helps alleviate the forgetting up to a certain degree, repeated training on the memory samples can lead to overfitting on these [32, 54]. To address this, previous works [8, 55] exploit functional priors for regularizing the visual space alongside memory replay. Here, we propose to regularize the past task distributions in the textual space by using the hand-crafted prompt-based features $\{\mathbf{t}_y^{h,l}\}_{l=1}^L$ to distill the past task latent samples $\left\{ z^i = \{z_m^i\}_{m=1}^M \right\}_{i=1}^{t-1}$. Namely, the probability of the sample set $z^t$ belonging to a class $y \in C^t$ is:

$$P_{\text{KD}}(y|z^t) = \frac{1}{M} \sum_{m=1}^M \frac{1}{L} \sum_{l=1}^L \frac{\exp\left(\langle \mathbf{t}_y^{h,l}, z_m^t \rangle\right)}{\sum_{c=1}^{|C^t|} \exp\left(\langle \mathbf{t}_c^{h,l}, z_m^t \rangle\right)}. \tag{7}$$

The resulting language-aware distillation loss is thus the sum of cross-entropy between the true label distribution $y_c$ and the predicted probability distribution $P_{\text{KD}}$ across all the past-task classes:

$$\mathcal{L}_{\text{KD}} = - \sum_{t=1}^{T-1} \sum_{c=1}^{|C^t|} \log P_{\text{KD}}(c|z^t) y_c, \tag{8}$$

where $\mathcal{L}_{\text{KD}}$ serves as a **data-free** (*i.e.*, no training samples required) text-to-text distribution regularizer that encourages the latent variable outputs from past-task adapters to stay close to the text features from the hand-crafted prompts. $\mathcal{L}_{\text{KD}}$ is applied only during the memory consolidation training, that is, when the past-task adapters are trainable. Lastly, as $\mathcal{L}_{\text{KD}}$ acts on the functional space of past tasks, this sets apart our setting from the non-CL setup of [56] where the language-aware distillation loss regularizes the vector embedding space.

### 3.3.2 Task-specific adapter initialization considering stability

Stability gap [57] in CL refers to the temporary yet substantial forgetting of past task knowledge in the initial phases of updating a network's weights to learn an incremental task. An informed weight initialization can help bridge this gap over random initialization by stabilizing the learning for new task components [58]. We thus leverage the $t-$th task text features $\{\mathbf{t}_y^{h,l}\}_{l=1}^L$ to initialize the weights $\mathbf{w}_t^\mu, \mathbf{w}_t^\sigma \in \mathbb{R}^{d \times d}$ of our $t-$th task's linear adapter layer. Let $\mathbf{s}_\mu, \mathbf{s}_\sigma \in \mathbb{R}^{|C^t| \times d}$ be the mean and the std. dev. of the $L$ text features. We initialize $\mathbf{w}_t^\mu$ and $\mathbf{w}_t^\sigma$ as:

$$\mathbf{w}_t^\mu = \frac{1}{d} \langle \mathbf{s}_\mu^T, \mathbf{s}_\mu \rangle, \quad \mathbf{w}_t^\sigma = \frac{1}{d} \langle \mathbf{s}_\sigma^T, \mathbf{s}_\sigma \rangle. \tag{9}$$

## 3.4 Training objective

**Approximate ELBO.** Building upon Eq. (4), we now learn the task-specific adapters $q_\phi^t$ to approximate the intractable $t \in [1, T]$ task-specific posteriors. The ELBO (see App. F for derivation) is:

$$\log p(y^{1:T}|\mathbf{x}; \tilde{\mathbf{t}}_c^t) \geq \sum_{t=1}^T \left[ \mathbb{E}_{q_\phi^t(z^t|\mathbf{x}; \tilde{\mathbf{t}}_c^t)} \left[ \log p_\theta(y^t|z^t, \mathbf{x}; \tilde{\mathbf{t}}_c^t) \right] - \mathbb{D}_{\text{KL}}\left( q_\phi^t(z^t|\mathbf{x}; \tilde{\mathbf{t}}_c^t) \| p_\chi(z^t) \right) \right]. \tag{10}$$

**Overall objective.** Denoting the loss weights by $\lambda$ and $\gamma$, our total loss term can be given as $\mathcal{L} = \mathcal{L}_{\text{CE}} - \lambda \mathbb{D}_{\text{KL}} + \gamma \mathcal{L}_{\text{KD}}$, where the cross-entropy $\mathcal{L}_{\text{CE}}$ and the prior-matching $\mathbb{D}_{\text{KL}}$ terms act on the outputs of all task encoders while the distribution regularization term $\mathcal{L}_{\text{KD}}$ acts only on the past task encoders. $\lambda$ is set to 0.001. As the past task encoders are trainable only during the memory consolidation training stage, $\lambda$ for these is set to 0 during training. $\gamma$ is set to 15.

## 4 Experiments

**Datasets.** We evaluate our method on CIFAR100 [3, 30], ImageNet100 [41, 43], ImageNet-R [59], CUB200 [60], and VTAB [60]. CIFAR100 [61] and ImageNet100 [62] setups split their respective original datasets into 10 tasks with 10 classes each. ImageNet-R [63] and CUB200 split 200 classes

| Method | CIFAR100 | | ImageNet100 | | ImageNet-R | | CUB200 | | VTAB | |
|---|---|---|---|---|---|---|---|---|---|---|
| | Avg ↑ | Last ↑ | Avg ↑ | Last ↑ | Avg ↑ | Last ↑ | Avg ↑ | Last ↑ | Avg ↑ | Last ↑ |
| Single-task JOINT | 80.28 | | 81.08 | | 80.92 | | 75.4 | | 89.29 | |
| Task-specific JOINT | 82.9 | | 83.55 | | 83.07 | | 85.72 | | 94.6 | |
| iCaRL [30] | 72.93 | 57.6 | 68.62 | 59.5 | 66.34 | 43.71 | 82.39 | 75.1 | 53.38 | 41.6 |
| L2P [65] | 78.92 | 70.04 | - | - | 77.07 | 69.33 | 76.98 | 68.47 | - | - |
| DualPrompt [59] | 82.11 | 74.31 | - | - | 82.73 | 76.41 | 82.37 | 76.29 | - | - |
| CODA-P [38] | 85.19 | 76.4 | 85.93 | 79.02 | 82.06 | 79.5 | 84.77 | 80.39 | 87.5 | 81.2 |
| PROOF [39] | 84.84 | 76.55 | - | - | 84.89 | 79.7 | 83.98 | 79.35 | - | - |
| Continual-CLIP [14] | 78.65 | 68.26 | 83.99 | 74.2 | 84.43 | 76.94 | 67.0 | 54.8 | 68.5 | 60.97 |
| CoOp [12] | 81.17 | 70.58 | 79.14 | 64.9 | 84.7 | 78.66 | 76.62 | 68.53 | 87.06 | 81.25 |
| MaPLe [20] | 82.74 | 74.52 | 79.23 | 64.06 | 85.28 | 79.71 | 73.38 | 64.43 | 83.91 | 81.81 |
| AttriCLIP [19] | 79.31 | 68.45 | 82.29 | 70.76 | 83.09 | 76.53 | 65.26 | 52.12 | 71.84 | 64.09 |
| CLIP-Adapter [13] | 78.75 | 68.32 | 84.13 | 73.96 | 84.49 | 78.1 | 67.41 | 54.49 | 68.23 | 61.02 |
| VPT [17] | 73.4 | 59.33 | 80.51 | 61.09 | 81.66 | 74.0 | 69.14 | 60.03 | 67.2 | 77.26 |
| Ours w/o VI | 84.36 | 76.8 | 86.11 | 76.48 | 85.69 | 79.83 | 72.21 | 61.87 | 90.74 | 88.64 |
| Ours | **86.13** | 78.21 | **87.76** | 79.16 | 85.77 | 79.98 | 86.93 | 81.64 | 91.37 | 89.67 |
| CoOp + Ours | 85.71 | 77.4 | 86.8 | 78.18 | 85.32 | 79.52 | **86.99** | **81.95** | **92.51** | **91.28** |
| MaPLe + Ours | 86.06 | **78.48** | 87.47 | 79.02 | 86.25 | 80.56 | 81.53 | 74.24 | 90.97 | 88.83 |
| AttriCLIP + Ours | 78.06 | 67.59 | 87.37 | **79.3** | **86.35** | **80.6** | 83.71 | 79.01 | 74.84 | 71.12 |

Table 1: **Performance comparison** of different methods averaged over three runs. Best scores are in **bold**. The second-best scores are in blue. The results for L2P, DualPrompt, and PROOF are taken from [39]. See App. Table 12 for statistical significance of these results using std. dev. scores.

into 10 tasks with 20 classes each. VTAB has 5 tasks with 10 classes each [64]. While CIFAR100, ImageNet100, and CUB200 are robust settings for evaluating CL methods in the face of large forgetting, ImageNet-R and VTAB make challenging settings for CL methods using pre-trained models as these might include test images in their pre-training set (see App. A.1 for details).

**Baselines.** We compare CLAP4CLIP against several baselines and state-of-the-art finetuning methods. These include: (a) CLIP-based methods – Continual-CLIP [14], CoOp [12], CLIP-Adapter [13], AttriCLIP [19], MaPLe [20], and PROOF [39], (b) vision-only methods – DualPrompt [59] L2P [65], CODA-P [38], (c) the baseline CIL method – iCaRL [30]. For a fair comparison, we adhere to the experimental protocols of PROOF [39] throughout. We adopt ViT-B/16 with the pre-trained weights of OpenAI [1] as our backbone unless otherwise specified. As the upper bounds on performance, we use the CLAP4CLIP with single and task-specific encoders, trained on all tasks jointly (JOINT).

**Variants.** We integrate our method with four prompt-based approaches: Ours uses CLAP with hand-crafted prompt templates, CoOp + Ours with soft prompts [12], MaPLe + Ours uses multi-modal soft prompts [20], and AttriCLIP + Ours uses CLAP4CLIP with instance-conditioned soft prompts [19]. Ours w/o Variational Inference (VI) is the deterministic variant of Ours depicted in App. Fig. 7. We leave the details of the training and the hyperparameters of our models in App. A.2.

**Performance measures.** To quantify CL performances, we report: (a) the final accuracy after the last incremental step (Last) and the average of the accuracies after each step (Avg) [30], and (b) the backward transfer score (BwT) [66] to quantify forgetting. To assess the benefits of probabilistic modelling, we report: (a) the expected calibration error (ECE) [67] that measures the calibration in the model's predictions [68], and (b) the (forward) transfer score [69] that quantifies the generalizability of CL models by measuring the extent of their zero-shot transfer ability after finetuning.

### 4.1 Results

**Accuracy.** We report performances in Table 1 on all five datasets. Our method consistently achieves the best results among all the methods compared. Notably, on CIFAR100 and ImageNet100, our variants using the hand-crafted and multi-modal prompts outperform the others. On the challenging ImageNet-R setup with significant intra-class diversity, our method can better leverage the instance-conditioned prompt knowledge of AttriCLIP [19], which helps it outperform PROOF [39] by $1.46\%$ in terms of average accuracy. On CUB200 and VTAB, sharing the prompt pool among all tasks gives CoOp [12] an edge over other baselines. Leveraging CoOp offers us the best results on these while surpassing PROOF, which also builds upon CoOp with task-specific soft prompts. We also observe that VPT [17] lags on all CL settings. Comparing the performance evolution of our variants against other baselines shows that our variants perform better throughout the incremental steps (App. Fig. 8).

**Forgetting.** Table 13 shows that in general, plugging CLAP4CLIP with prompt-based finetuning methods helps improve the BwT scores of the latter. It is worth noting that on the cross-dataset setting of VTAB [64], our variants are the only methods that effectively transfer the knowledge learned from

| Method | ImageNet100 | CIFAR100 + ImageNet100 |
|---|---|---|
| DualPrompt [59] | 81.9 | 67.1 |
| Continual-CLIP [14] | 75.4 | 54.9 |
| CoOp [12] | 79.3 | 55.4 |
| MaPLe [20] | 84.81 | 76.2 |
| AttriCLIP [19] | 83.3 | 78.3 |
| PROOF [39] | 81.26 | 82.59 |
| Ours | 83.51 | 83.83 |
| CoOp + Ours | 82 | 82.63 |
| MaPLe + Ours | 82.97 | 83.6 |
| AttriCLIP + Ours | **84.14** | **84.56** |

Table 2: **Performance comparison on the CDCL setting** [19]. All CLIP-based methods use the ViT-L/14 backbone.

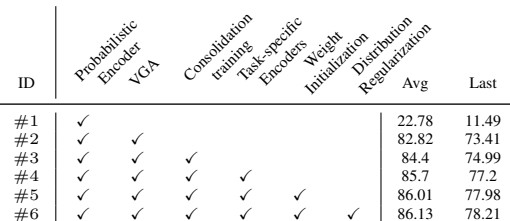

| ID | Probabilistic Encoder | VGA | Consolidation training | Task-specific Encoders | Weight Initialization | Distribution Regularization | Avg | Last |
|---|---|---|---|---|---|---|---|---|
| #1 | ✓ | | | | | | 22.78 | 11.49 |
| #2 | ✓ | ✓ | | | | | 82.82 | 73.41 |
| #3 | ✓ | ✓ | ✓ | | | | 84.4 | 74.99 |
| #4 | ✓ | ✓ | ✓ | ✓ | | | 85.7 | 77.2 |
| #5 | ✓ | ✓ | ✓ | ✓ | ✓ | | 86.01 | 77.98 |
| #6 | ✓ | ✓ | ✓ | ✓ | ✓ | ✓ | 86.13 | 78.21 |

Table 3: **Ablations** of the key components of CLAP4CLIP on CIFAR100.

incremental tasks to improve the performance on past tasks (*i.e.*, BwT > 0). This indicates that our probabilistic modeling strategy does not only counter forgetting but can also help bring anti-forgetting properties onto existing finetuning methods.

**Calibration.** App. Table 15 compares the ECE scores of our variants and their respective underlying deterministic baselines at the last test step. In general, our variants help enhance (decrease) the ECE scores of the underlying prompt-based methods. This implies that even in the face of forgetting in a CL setup, CLAP retains more reliability in assessing the confidence of its predictions.

**Generalization.** App. Table 14 shows that our method consistently enhances the (forward) transfer scores of the underlying deterministic prompt-based methods. This means that CLAP can better transfer the learned knowledge from seen tasks to help solve future tasks.

**Resource-constrained CL.** To study the robustness of CLAP towards memory and compute-constrained environments, we ablate its performance on *replay-free* [70] and *computationally-budgeted* [23] CL setups, respectively. Tables 16 and 17 show that for both these setups, leveraging the instance-conditioned and semantically diverse prompts of AttriCLIP provides an edge. Here, our variant leveraging AttriCLIP surpasses the replay-free SOTA, *i.e.,* CODA-P [38] and the budgeted SOTA, *i.e.,* AttriCLIP [19]. Further ablating the role of our proposed language-aware distribution regularization and weight initialization components for our AttriCLIP variant shows that the former component remains crucial for avoiding forgetting under resource-constrained settings.

### 4.1.1 Cross-Datasets Continual Learning (CDCL)

To simulate real-world settings with long sequence of tasks and large distribution shifts, the CDCL setting [19] trains a model sequentially on ImageNet100 and CIFAR100 (*i.e.*, on 20 tasks), and evaluates it jointly on these. For a fair comparison with [19], we adopt the ViT-L/14 as our CLIP backbone and set the train/test batch size to 32. All other settings remain the same as in Sec. 4.1. Table 2 reports the last task accuracy of different methods. While all our variants improve the CDCL performances of their respective baselines, combining ours with AttriCLIP [19] leads to the most gains. This further suggests that our framework can reliably leverage the diverse nature of learned prompts to inherit their setting-specific advantages.

### 4.2 Ablation Studies

We provide a few ablations of the training pipeline for CLAP4CLIP below and leave more in App. C.

**Influence of components.** We ablate the importance of different components of CLAP4CLIP in Table 3. On top of the base CLIP model, we first train a probabilistic encoder. Adding the VGA module and the memory consolidation training stage helps us achieve more stable performances while countering forgetting. We then apply task-specific encoders which make the centroids of the class-specific latent variable more separable (see Fig. 4) thus improving the last task accuracy by 2.21%. Language-aware weight initialization and regularization help improve the last task accuracies by 0.78% and 0.23%, respectively. Weight initialization further helps us tackle the *stability gap* [57, 58] (see App. C.6 for more ablations on language-aware components).

**Probabilistic vs Deterministic inference.** To understand our probabilistic inference modules further, we examine their performance against the deterministic variant of ours (Ours w/o VI). Table 1 shows that our probabilistic variant consistently outperforms its deterministic counterpart. This emphasizes the advantages of considering uncertainty in finetuning. We further introspect the effects of the number of layers for the VGA and task encoder modules in our framework in App. C.3.

**Time analyses.** We compare the inference time per iteration for different methods. As shown in App. Table 19, our variants need more inference time than other finetuning methods for the performance gains. The increased time comes mainly from the VGA and from inferring the $M$ latent variables.

**Parameter analyses.** The additional parameters in CLAP4CLIP come from the shared VGA module and the task-specific encoders. For a ViT-B/16 backbone of output dimension, $d = 512$ on CIFAR100, the VGA module contains 4,204,032 parameters. The mean and the std. dev. layers for 10 tasks have $d \times d$ parameters each, *i.e.*, $524, 2880$ parameters. Hence, the CLAP4CLIP has 9.5 million extra parameters, which is negligible compared to the pre-trained CLIP with $\approx 150$ million parameters. We report the parameter counts in Fig. 5.

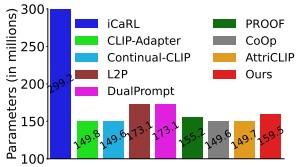

Figure 5: Parameter count comparison.

## 5 Out-of-the-box utilities of probabilistic finetuning

We study the out-of-the-box utilities of CLAP4CLIP's uncertainty quantification (UQ) capabilities. Our motivation for these is not to achieve state-of-the-art performance but to highlight the perks of probabilistic modeling in scenarios where the deterministic CL finetuning methods struggle.

**Post-hoc novel data detection (PhNDD).** PhNDD uses a pre-trained classification model to identify novel data based on the output confidence [71, 72]. For CL, this can help discern the arrival of new tasks, expand the network, etc. To evaluate the PhNDD capabilities of models within a CL setup, we design a simple setting. Namely, at all but the last test step, we treat the test data from the past and the current tasks as *seen* while those from all future tasks as *novel*. We then use FPR95, AUROC [73], and AUPR [74] scores as

| Method | AUROC ↑ | AUPR ↑ | FPR95 ↓ |
|---|---|---|---|
| Continual-CLIP [14] | 74.46 | 71.11 | 77.33 |
| Ours w/o VI | **82.29** | 78.88 | 68.83 |
| Ours | 82.21 | **79.54** | **68.72** |
| CoOp [12] | 80.15 | 77.62 | 66.8 |
| + Ours w/o VI | 81.98 | 78.88 | 66.21 |
| + Ours | **83.73** | **80.97** | **62.68** |

Table 4: PhNDD performances averaged over 3 runs on CIFAR100. Best scores for each variant are in **bold**.

our performance metrics (see App. D.1) averaged over all but the last incremental test steps. To quantify the output confidence, we rely on the Energy score [75] given its aptness for pre-trained models. Table 4 compares the averaged PhNDD performances. Our probabilistic models enhance the PhNDD capabilities of their underlying prompting frameworks. Moreover, the inferior results of the deterministic (*i.e.*, w/o VI) versions of our models suggest that probabilistic modelling helps a model output predictions that better express what it is not aware of.

**Exemplar selection.** We employ the entropy (averaged over $M$ predictions) of CLAP's softmax outputs as our exemplar selection criteria [2]. Table 20 shows the efficacy of entropy-based rehearsal for our method, where other deterministic methods lag due to their inconsistent UQ capabilities. Next, we employ the energy [75] and the variance of the softmax outputs as our selection criterion and contrast these against other criteria proposed in [2]. Fig. 6 shows that variance-based exemplar selection outperforms random, and is only second to iCaRL [30] in terms of Last accuracy. We note that deterministic methods with pointwise predictions cannot use variance for exemplar selection.

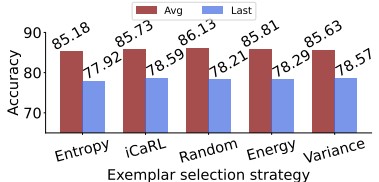

Figure 6: **Analyses of various strategies** for exemplar selection w/ our method on CIFAR100.

## 6 Conclusion

In this paper, we propose CLAP4CLIP, a probabilistic finetuning method for learning task-specific distributions over visual-guided textual features. Our model shares the visual-guided text alignment module across all tasks while adding lightweight task-specific encoders to learn fine-grained task distributions. Besides leading to little memory overhead, this architecture is compatible with several prompt-tuning-based methods thus helping us inherit their respective perks on different CL settings. Our experiments show the superior results of CLAP4CLIP across several datasets and settings. We conclude with two out-of-the-box utilities of our method wherein existing continual learning methods lag: post-hoc novel data detection and uncertainty-based exemplar selection. We discuss our limitations, potential future research directions, and the broader impact in App. sec. E.

# 7    Acknowledgement

This work was partially supported by a Discovery Early Career Researcher Award Fellowship (DE230101591) awarded by the Australian Research Council (ARC) to Dong Gong. We are grateful to Daniel Marczak and M. Jehanzeb Mirza for their insights on the need for continual finetuning of the pre-trained CLIP model.

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

# Appendix

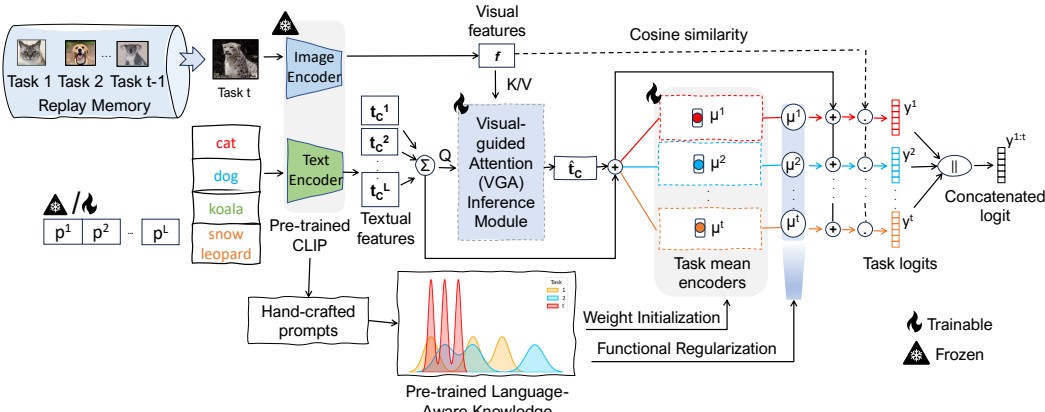

Figure 7: **Illustration of the deterministic variant of Ours (Ours w/o VI in Table 1):** the task-specific text features are fed to their respective task encoders, consisting of only the mean $\mu$ layer each. There is no sampling involved and the task mean outputs are fused directly with the original task features prior to deriving the task logits $\mathbf{y}^t$. All task logits are concatenated to produce the final prediction $\mathbf{y}^{1:t}$.

## A  Experiments and Benchmarks

### A.1  Datasets

| Dataset | # training instances | # testing instances | # Classes | # Tasks | Link |
|---|---|---|---|---|---|
| CIFAR100 | 50,000 | 10,000 | 100 | 10 | URL |
| ImageNet100 | 130,000 | 5,000 | 100 | 10 | URL |
| ImageNet-R | 24,000 | 6,000 | 200 | 10 | URL |
| CUB200 | 9,430 | 2,358 | 200 | 10 | URL |
| VTAB | 1,796 | 8,619 | 50 | 5 | URL |

Table 5: **Benchmark datasets** and their details.

We evaluate our method on five datasets, the details of which are reported in Table 5. Following [39], we shuffle the order of training classes for all but the VTAB dataset with the random seed 1993. While the original VTAB [76] includes 19 evaluation tasks from three categories (*natural*, *specialized*, and *structured*) and their respective sub-domains, we rely on the five datasets cross-domain class-incremental subset proposed in SimpleCIL [64]. The five datasets (used in the same streaming order) include Resisc45 [77], DTD [78], Pets [79], EuroSAT [80], and Flowers [81]. To make the classes emerge from domain to domain, we do not shuffle the class order for VTAB.

#### A.1.1  Exemplar selection for memory replay

Following [30, 41], we employ the herding algorithm [44] to choose the exemplars for our main experiments. Following the previous works [82, 39], we rely on two typical methods to populate the memory:

1. **Fixed memory budget** maintains a static memory $\mathcal{M}$ with $K$ instances. Upon having seen $|\mathcal{Y}_b|$ number of classes after an incremental training stage, the model selects $\frac{K}{|\mathcal{Y}_b|}$ exemplars per class.

2. **Expandable exemplar set** dynamically expands the memory $\mathcal{M}$ with the arrival of more incremental tasks. After each incremental training stage, the model here stores $|\mathcal{Y}_b| \times k_c$ exemplars, where $k_c$ is the number of exemplars per class.

For CIFAR100, ImageNet100, and VTAB, given their lesser number of classes, we employ the first policy, and keep a total of 2,000, 1,000, and 1,000 exemplars, respectively. This amounts to the respective sub-totals of 20 and 10 exemplars per class after the last incremental stage. We choose these sizes for a straightforward comparison with the existing works, *i.e.,* PROOF [39] for CIFAR100 and AttriCLIP [19] for ImageNet100. For VTAB, the chosen memory size reflects the fact that we have only $1,796$ training instances in total (see Table 5). For ImageNet-R and CUB200 with 200 classes each, we adopt the second policy and store 20 exemplars per class.

## A.2 Training and Hyperparameter selection

We train CLAP and its variants using SGD, with a batch size of 64, for 5 epochs, including 1 epoch of linear warmup. The initial learning rate (LR) is set to 1e-3 and decays with cosine annealing. At the end of each incremental task ($t > 1$), we perform memory consolidation training for 2 epochs, with an LR of 1e-4, on the class-balanced memory dataset. All our experiments were performed on NVIDIA V100 GPUs hosted on the Gadi supercomputers of the National Computational Infrastructure (NCI Australia).

**Training for memory consolidation.** To alleviate the forgetting of past tasks, we finetune on the class-balanced dataset of new data and rehearsal data $\mathcal{M}$ at the end of each incremental training step ($t > 1$) [41, 27]. Following other well-established parameter-isolation CL algorithms [27, 42, 52], we freeze the past task encoders during the normal training. This helps us avoid knowledge interference from the dominant new task training samples. During the memory consolidation training stage, we optimize all the task encoders while freezing the task-shared VGA module parameters.

**Hyperparameter tuning.** To the end goal of obtaining task-agnostic hyperparameters [83], we tuned our hyperparameters using a validation set comprising $10\%$ of the CIFAR-100 training dataset. Similar to [27], performing the hyperparameter search only on the CIFAR100 setup helps us avoid optimizing for the number of tasks while generalizing across all our other setups. Table 6 shows the candidate values for the hyperparameter grid search and their best-chosen values. Tables 7, 8, 9, and 10 report the last task accuracy scores (Last) corresponding to the hyperparameter search for the number of training epochs, the number of finetuning epochs, the coefficient $\gamma$ and the coefficient $\lambda$, respectively. Fig. 9 in the main paper reports the accuracy and the runtimes for the different numbers of MC samples $M$. *We will release the full source code upon the acceptance of our paper.*

| Hyperparameter | Range | Chosen value |
|---|---|---|
| Learning rate | $5e$-3, $1e$-3, $5e$-4 | $1e$-3 |
| Epochs | $3, 5, 7$ | 5 |
| Warmup epochs | $0.5, 1, 1.5$ | 1 |
| Finetuning epochs | $1, 2, 3, 4$ | 2 |
| $\gamma$ | $1, 5, 10, 15, 20, 25$ | 15 |
| $\lambda$ | $0.0001, 0.001, 0.01, 0.1$ | 0.001 |
| $M$ | $1, 5, 10, 15, 20, 25, 30, 50$ | 20 |

Table 6: **Hyperparameter tuning**: we run a gridsearch on the CIFAR100 setup with a validation set comprising $10\%$ of the training set. The chosen values are reused across all other setups.

| Epochs | 3 | 5 | 7 |
|---|---|---|---|
| Last | 77.32 | 78.21 | 78.18 |

Table 7: Accuracy vs. Training epochs

| Finetuning ep. | 1 | 2 | 3 | 4 |
|---|---|---|---|---|
| Last | 77.65 | 78.21 | 78.2 | 78.18 |

Table 8: Accuracy vs. Finetuning epochs

| $\gamma$ | 1 | 5 | 10 | 15 | 20 | 25 |
|---|---|---|---|---|---|---|
| Last | 78.04 | 77.94 | 78.1 | 78.21 | 77.96 | 77.14 |

Table 9: Accuracy vs. weight "$\gamma$" for $\mathcal{L}_{KD}$

| $\lambda$ | 0.0001 | 0.001 | 0.01 | 0.1 |
|---|---|---|---|---|
| Last | 78.16 | 78.21 | 77.99 | 77.4 |

Table 10: Accuracy vs. weight "$\lambda$" for $\mathbb{D}_{KL}$

### A.3 Variational modeling of text feature space vs. image feature space

We opt for the probabilistic modeling of task-specific text feature space rather than the image feature space mainly in light of the practical constraints imposed by the class-incremental learning (CIL) setting. In CIL, at test time, we are not given the task labels for images. As such, if we were to use task-specific adapters to model task-specific visual features distribution (rather than task-specific text features distribution), then we must infer which images are to be routed to what adapter. Existing solutions [8] to task id inference would route the text-guided visual features to all available adapters and then infer the correct prediction based on the adapter's outputs. Such an exhaustive routing mechanism greatly increases the test-time computational burden. Instead, we exploit the multimodal nature of CLIP [1] to model the distribution of visual-guided text features. This helps us avoid test-time task id inference as now our visual features form a shared context to which all task-specific text features (which we can distinguish simply by their labels) can attend. By sampling from the distributions over such visual-guided task-specific text features, we compute their cosine similarities with the visual features to obtain our predictive logits.

### A.4 Latency comparison for VT-CLIP styled VGA vs Ours

We compare the performance of VT-CLIP-styled VGA with Ours. To align the text features with the image features, the former uses per-pixel spatial features obtained from the ViT prior to global pooling while we use the globally pooled features. Table 11 shows that VT-CLIP styled VGA achieves similar accuracy as ours while incurring $\approx 6\times$ higher inference time.

| Method | Avg. | Last | Inference time (s) |
|---|---|---|---|
| VT-CLIP styled VGA | **86.54** | 77.98 | 0.94 |
| Ours | 86.13 | **78.21** | **0.16** |

Table 11: Performance comparison of VT-CLIP styled VGA with Ours on CIFAR-100.

### A.5 Algorithm overview.

## B Results

### B.1 Performance evolution

To complement the results in Table 1, Fig. 8 compares the accuracy of different methods at each evaluation step across all datasets. Our major conclusions are briefed as follows. A) The base task performance of CLAP4CLIP (ours) is consistently higher than other methods including the state-of-the-art PROOF [39]. This suggests that our probabilistic finetuning framework is effective for general downstream tasks in a non-incremental setting. B) For the CL settings in Table 1 where either of the CLAP4CLIP variants achieve the best performances, their performance curves also consistently retain superior results across all evaluation steps. This validates the effectiveness of our method at tackling forgetting. C) Similar to [39], we notice that CLAP4CLIP achieves a significant performance improvement over vision-only methods (L2P and DualPrompt). This indicates the merits of considering text and visual cues together for continual learning.

**Algorithm 1:** A forward CLAP4CLIP pass at test step $t$

**Input** : $\{\mathbf{t}^i\}_{i=1}^t$: text features, $f(x)$: image features
**Output** : $\hat{y}^{1:t}$ (predictions for classes seen till task $t$)

1   $\{\hat{\mathbf{t}}^i\}_{i=1}^t \leftarrow \text{VGA}(\{\mathbf{t}^i\}_{i=1}^t, f(x))$        // Eq. (5a)
2   **for** $i \leftarrow 1;\ i \le t;\ i \mathrel{+}= 1$ **do**
3      $\tilde{\mathbf{t}}^i = \hat{\mathbf{t}}^i + \mathbf{t}^i$        // Eq. (5b)
4      $\mathcal{N}(\mu^i, \sigma^i) \leftarrow q_\phi^i(\tilde{\mathbf{t}}_c^i)$        // Sec. 3.2.3
5      $\hat{y}^i \leftarrow \emptyset$        // Null prediction set
6      **for** $m \leftarrow 1;\ m \le M;\ m \mathrel{+}= 1$ **do**
7          $z_m^i \sim \mathcal{N}(\mu^i, \sigma^i)$        // Sampling
8          $\tilde{\mathbf{t}}_m^i \leftarrow \tilde{\mathbf{t}}^i + z_m^i$        // Fusion
9          $\hat{y}_m^i \leftarrow \langle f(\mathbf{x})^T, \tilde{\mathbf{t}}_m^i \rangle$        // Using Eq. (3b)
10         $\hat{y}^i \leftarrow \hat{y}^i \cup \hat{y}_m^i$        // Set Union
11   $\hat{y}^{1:t} \leftarrow [\hat{y}^1, ..., \hat{y}^t]$        // Concatenation

| Method | CIFAR100 | ImageNet100 | ImageNet-R | CUB | VTAB |
|---|---|---|---|---|---|
| Continual-CLIP [14] | 1.416 | 2.175 | 1.98 | 2.087 | 0.614 |
| +Ours | 1.39 | 2.19 | 1.86 | 2.06 | 0.443 |
| CoOp [12] | 1.57 | 2.47 | 1.95 | 1.99 | 0.54 |
| +Ours | 1.533 | 2.074 | 2.011 | 1.885 | 0.516 |
| MaPLe [20] | 1.3 | 2.052 | 2.16 | 1.803 | 0.49 |
| +Ours | 1.36 | 1.956 | 1.84 | 1.62 | 0.407 |
| AttriCLIP [19] | 1.781 | 2.54 | 2.37 | 2.419 | 0.996 |
| +Ours | 1.677 | 2.019 | 2.388 | 2.410 | 0.98 |

Table 12: **Standard deviation** (std. dev.) scores comparison for Avg. accuracy scores of Table 1 between our variants and their corresponding baseline prompt-based finetuning methods over three runs. In general, our std. dev. scores are comparable to or lower than the corresponding baseline methods and are thus statistically significant.

| Method | CIFAR100 | ImageNet100 | ImageNet-R | CUB | VTAB |
|---|---|---|---|---|---|
| Continual-CLIP [14] | **-0.086** | **-0.091** | **-0.066** | -0.124 | -0.041 |
| +Ours | -0.106 | -0.117 | -0.107 | **-0.117** | **0.012** |
| CoOp [12] | -0.257 | -0.338 | -0.12 | -0.162 | -0.007 |
| +Ours | **-0.129** | **-0.139** | **-0.112** | **-0.106** | **0.011** |
| MaPLe [20] | -0.209 | -0.352 | -0.1 | -0.145 | **0.037** |
| +Ours | **-0.105** | **-0.112** | **-0.093** | **-0.102** | 0.005 |
| AttriCLIP [19] | **-0.128** | -0.152 | **-0.082** | -0.151 | -0.099 |
| +Ours | -0.143 | **-0.1** | -0.092 | **-0.037** | **0.041** |

Table 13: **Backward Transfer** (BwT) scores ↑ comparison between our variants and their corresponding baseline prompt-based finetuning methods averaged over three runs. Best scores across each pair is highlighted in **bold**.

| Method | CIFAR100 | ImageNet100 | ImageNet-R | CUB | VTAB |
|---|---|---|---|---|---|
| Continual-CLIP [14] | 65.34 | **53.13** | 61.67 | **59.55** | 65.13 |
| +Ours | **65.47** | 53.07 | **64.05** | 58.11 | **66.91** |
| CoOp [12] | 64.09 | 52.6 | 60.93 | **62.11** | 69.38 |
| +Ours | **66.2** | **55.09** | **63.44** | 58.6 | **74.1** |
| MaPLe [20] | 68.22 | 57.04 | 66.56 | 61.6 | 71.51 |
| +Ours | **76.17** | **62.33** | **70.03** | **67.8** | **78.29** |
| AttriCLIP [19] | 61.45 | 50.4 | 56.41 | 57.04 | 61.59 |
| +Ours | **61.87** | **50.56** | **58.03** | **57.95** | **64.3** |

Table 14: **Transfer** scores [84] ↑ comparison between our variants and their corresponding baseline prompt-based finetuning methods averaged over three runs. Best scores across each pair is highlighted in **bold**.

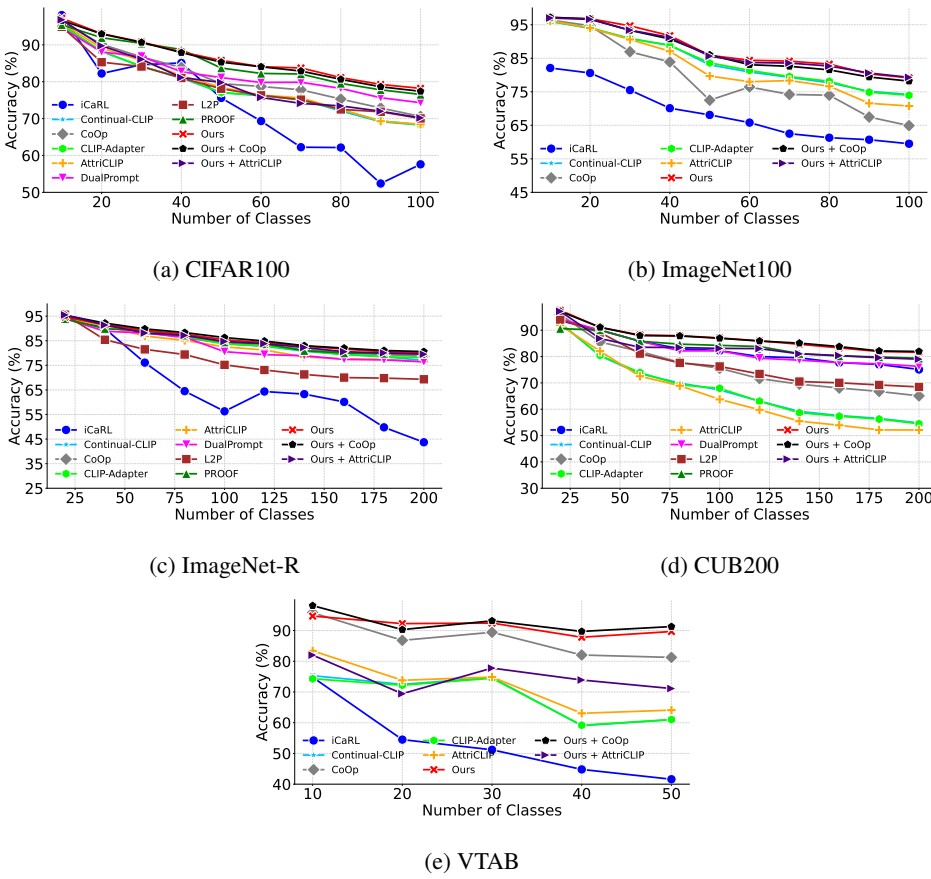

(a) CIFAR100

(b) ImageNet100

(c) ImageNet-R

(d) CUB200

(e) VTAB

Figure 8: **Performance evolution of different methods.** The top-1 accuracy (%) is reported upon learning of each task.

| Method | CIFAR100 | ImageNet100 | ImageNet-R | CUB | VTAB |
|---|---|---|---|---|---|
| Continual-CLIP [14] | 0.288 | 0.238 | 0.206 | 0.208 | 0.186 |
| +Ours | **0.216** | **0.207** | **0.201** | **0.203** | **0.165** |
| CoOp [12] | 0.245 | 0.3 | **0.191** | 0.21 | 0.191 |
| +Ours | **0.224** | **0.217** | 0.207 | **0.204** | **0.136** |
| MaPLe [20] | **0.168** | 0.243 | 0.149 | 0.195 | 0.195 |
| +Ours | 0.214 | **0.208** | **0.146** | **0.184** | **0.159** |
| AttriCLIP [19] | **0.256** | 0.256 | 0.205 | 0.209 | **0.191** |
| +Ours | 0.304 | **0.205** | **0.19** | **0.198** | 0.304 |

Table 15: **Expected Calibration Error** (ECE) scores ↓ (computed over 15 bins) comparison between our variants and their corresponding baseline prompt-based finetuning methods averaged over three runs. Best scores across each pair is highlighted in **bold**.

## B.2 Results for replay-free CL setup

| Method | CIFAR100 | | ImageNet100 | | ImageNet-R | | CUB200 | | VTAB | |
|---|---|---|---|---|---|---|---|---|---|---|
| | Avg ↑ | Last ↑ | Avg ↑ | Last ↑ | Avg ↑ | Last ↑ | Avg ↑ | Last ↑ | Avg ↑ | Last ↑ |
| CODA-P | 74.66 | 63.7 | 79.8 | 72.66 | 76.44 | 72.19 | 74.32 | 70.15 | 70.59 | 66.12 |
| AttriCLIP [19] | 71.35 | 61.4 | 80.55 | 73.08 | 79.57 | 76.1 | 73.89 | 72.36 | 71.25 | 66.02 |
| CoOp + Ours | 74.19 | 63.45 | 81.07 | 72.0 | 81.22 | 75.8 | **82.59** | **74.98** | **82.11** | **80.11** |
| AttriCLIP + Ours | **76.94** | **69.39** | **84.1** | **75.83** | **85.2** | **78.57** | 77.2 | 73.58 | 72.98 | 68.5 |
| AttriCLIP + Ours (w/o adapter init) | 71.87 | 60.35 | 75.9 | 69.41 | 77.1 | 70.53 | 69.82 | 65.95 | 65.48 | 61.35 |
| AttriCLIP + Ours (w/o distribution reg.) | 64.2 | 53.01 | 64.03 | 52.55 | 62.81 | 53.19 | 58.52 | 51.0 | 58.22 | 56.14 |

Table 16: **Performance comparison without memory replay** (avg. over 3 runs). For a thorough analysis, the last two rows ablate our proposed pretrained CLIP's language-aware anti-forgetting components (Sec. 3.3, main paper): distribution regularization and adapter weight initialization. Best results are in **bold**.

## B.3 Results for computationally-budgeted CL setup

For our computationally-budgeted CL setup, we follow [23] where on each incremental training task, we allocate the number of training iterations equivalent to 1 epoch on the first (base) task of each dataset. Here, our variant utilizing instance-conditioned prompts of AttriCLIP outperforms other compared methods. A further ablation shows that our proposed weight distribution regularization technique indeed remains a crucial component at tackling forgetting on the budgeted setup (see the two bottom-most rows in Table 17).

| Method | CIFAR100 | | ImageNet100 | |
|---|---|---|---|---|
| | Avg ↑ | Last ↑ | Avg ↑ | Last ↑ |
| CODA-P | 52.13 | 49.5 | 52.99 | 48.03 |
| AttriCLIP [19] | 58.61 | 52.1 | 60.54 | 57.4 |
| PROOF | 55.29 | 50.3 | 56.8 | 54.37 |
| AttriCLIP + Ours | **61.7** | **55.89** | **62.91** | **60.2** |
| AttriCLIP + Ours (w/o init.) | 61.33 | 54.95 | 62.14 | 59.86 |
| AttriCLIP + Ours (w/o reg.) | 58.95 | 52.6 | 60.04 | 57.93 |

Table 17: **Results for computationally budgeted CL setup [23]:** we follow the "Normal" budget setup from [23] where each incremental task is allocated training iterations equivalent of 1 epoch on the first task of each dataset. Scores reported are averages over three runs. Best results are in **bold**.

# C Ablation studies

## C.1 Sensitivity to the number of Monte Carlo (MC) samples.

We vary the number of MC samples $M$ from 1 to 50. In Fig. 9, the accuracy is poorer in range $[1, 10]$, grows in range $[10, 20]$, and saturates thereafter. Hence, we set $M$ to 20 for all our experiments.

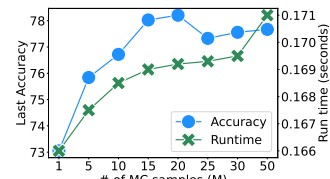

Figure 9: Accuracy-runtime trade-off with number of MC samples $M$.

## C.2 Effect of forgetting on individual task adapters.

We ablate the task head predictions over the test set of each task on CIFAR100 (see App. C.2 for more details). Fig. 10a reports the accuracy of test set samples corresponding to the task encoder heads at the end of incremental training on the last task. Here, the first row is to be interpreted as follows: $77\%$ of test samples belonging to test set of the first task (test set ID 1) were *correctly* allocated to task head 1, $1.4\%$ of test samples belonging to test set of the first task (test set ID 1) were *incorrectly* allocated to task head 2, and so on. Visualizing the task head selection results for the last task evaluation helps us uncover the amount of forgetting among the individual task heads at the end of the incremental training.

Fig. 10b compares the evolution of the task head selection accuracy across the incremental test steps. Here, at the first test step, we have only one task head and thus the task head selection accuracy is $100\%$. At the second test step, we have the test samples from two seen tasks as well as two available

task heads. Out of all test samples of task 1, the reported $94.5\%$ were correctly classified into the task head 1 while the rest $5.5\%$ were incorrectly classified into the task head 2. Similarly, for test samples belonging to task 2, $3.1\%$ were incorrectly classified into the task head 1 while the reported $96.9\%$ were correctly classified into the task head 2, and so on. Hence, by studying the task head selection per incremental step, we can investigate the trend of forgetting among the individual task heads.

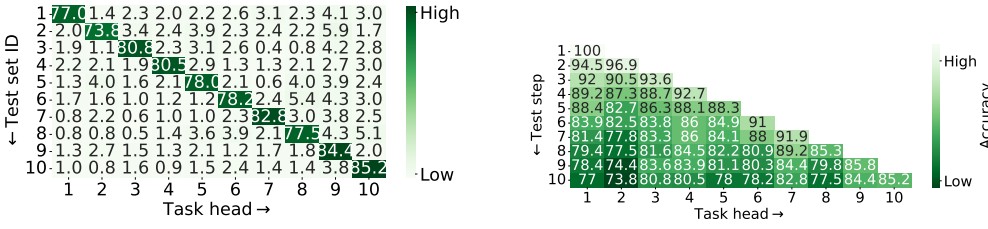

(a) Last step task head selection accuracies        (b) Per step task head selection accuracies

Figure 10: **Task head selection accuracies** reported on CIFAR-100 upon: (a) evaluation on the last step, (b) evaluation on each incremental step.

### C.3 Effect of inference module architecture.

To further investigate the effects of inference modules on performances, we vary the number of layers for the VGA module (sec. 3.2.2) and for the task-specific encoders (sec. 3.2.3). Fig. 11a reports the results of varying the number of Transformer Decoder layers [48] in the VGA module. As the number of layers grow, the average accuracy (Avg) increases while the last task accuracy (Last) decreases. This indicates that while a larger number of layers in the VGA module lead to an increase in the initial tasks' performances, these are amenable to larger forgetting on latter incremental steps.

In Fig. 11b, we report the performances for varying number of MLP layers in the mean and the standard deviation heads of the task distribution encoders. Unlike the VGA module, here we observe a consistent trend of decreasing last and average task accuracy with the increase in the number of layers. This clearly indicates the superiority of using a single-layered task distribution encoder.

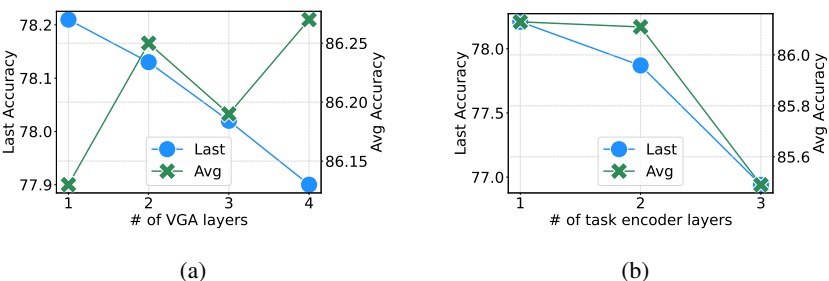

(a)                                 (b)

Figure 11: **Ablation studies on CIFAR100 showing:** (a) the variation of accuracy with the number of Transformer decoder layers in the VGA module, (b) the variation of accuracy with the number of linear layers in the task-specific mean and standard deviation encoders.

### C.4 Effect of prior type.

To study the role of a more informed prior in our VI framework, we study three choices of priors to be used in the prior-matching term of eq. (10): (a) the static (standard normal) prior, (b) the language-aware prior using the distribution obtained from the task encoders using the hand-crafted prompts' features $\{\mathbf{t}_y^{h,l}\}_{l=1}^{L}$ (sec 3.3), (c) the data-driven prior using a randomly chosen subset of a training minibatch as the context set to condition the prior on (see App. G for more details). App. Table 18 shows that while (b) and (c) slightly improve over (a) in terms of accuracies and forgetting, these come at the cost of poorer model calibration and longer runtime per iteration.

| Prior type | Last ↑ | Avg ↑ | BwT ↑ | ECE ↓ | Runtime per iter. ↓ |
|---|---|---|---|---|---|
| Static | 78.21 | 86.13 | -0.141 | **0.204** | **0.169** |
| Data-driven | 78.32 | 86.15 | -0.115 | 0.216 | 0.172 |
| Language-aware | **78.38** | **86.22** | **-0.112** | 0.214 | 0.17 |

Table 18: **Performances of different priors** averaged over 3 runs on CIFAR100.

## C.5 Inference time for different finetuning methods.

Table 19 investigates the inference time per iteration for different methods. Among the compared prompt-based methods, the inference time for AttriCLIP [19] is notably the highest. This is because it relies on selecting test instance-conditioned prompt tokens from a pool of prompt tokens. The instance-specific prompts are fed to the text encoder which further outputs an equivalent number of instance-specific text features to be used in the derivation of logits through eq. 1. These operations increase the inference time of AttriCLIP beyond our proposed variants of CLAP4CLIP with hand-crafted prompts (Ours), class-conditioned prompts (CoOp + Ours), and multi-modal prompts (MaPLe + Ours) where the latter three outperform AttriCLIP significantly across all our settings.

| Method | Inference time (s) |
|---|---|
| Continual-CLIP [14] | 0.017 |
| CoOp [12] | 0.018 |
| MaPLe [20] | 0.035 |
| AttriCLIP [19] | 0.257 |
| CLIP-Adapter [13] | 0.019 |
| Ours | 0.163 |
| CoOp + Ours | 0.182 |
| MaPLe + Ours | 0.064 |
| AttriCLIP + Ours | 0.299 |

Table 19: Average inference time for different finetuning methods on CIFAR100.

## C.6 Influence of language-aware knowledge components on training dynamics.

Continuing our ablations from sec. 4.2, here we visualize the effects of using language-aware pre-trained knowledge, *i.e.*, weight initialization and task distribution regularization on the training dynamics of our model. For thorough analyses, we consider four variants of our model: (a) Ours uses both weight initialization and task distribution regularization, (b) Ours without weight initialization, (c) Ours without task distribution regularization, and (d) Ours without either of the language-aware components.

**Does language-aware weight initialization help alleviate stability gap [57]?** To answer this, we first investigate the evolution of the training loss during the initial training stages of each incremental task. Fig. 14 shows the loss $\mathcal{L}$ (Sec. 3.4) during the initial 100 training iterations of each task. We observe that our proposed weight initialization technique leads to lower training losses for the scenarios with or without task distribution regularization, *i.e.*, in general, **red values** < **green values** and **blue values** < **orange values**. Following [58], our observations support the hypothesis that larger loss values lead to the **stability gap** [57] for CL, and that an informed weight initialization method can help tackle it by reducing the initial training loss.

To further verify the benefit of our proposed weight initialization strategy for reducing the stability gap, we ablate the accuracy evolution of the first task test samples during the early training stages of each task. Figures 12b and 12a contrast these for CIFAR100. In general, our proposed weight initialization strategy helps mitigate the drop in accuracy during the initial training phases. On average, the first task accuracy upon the first iteration of training across all tasks remains 78.12 without weight initialization and grows to 79.5 with weight initialization, *i.e.,* a gain of 1.38 percentage points.

**How does language-aware knowledge help learning of task distributions in general?** To understand the effect of language-aware knowledge on task distribution learning, we next investigate the evolution of the means and standard deviations learned by the past and the new task heads throughout

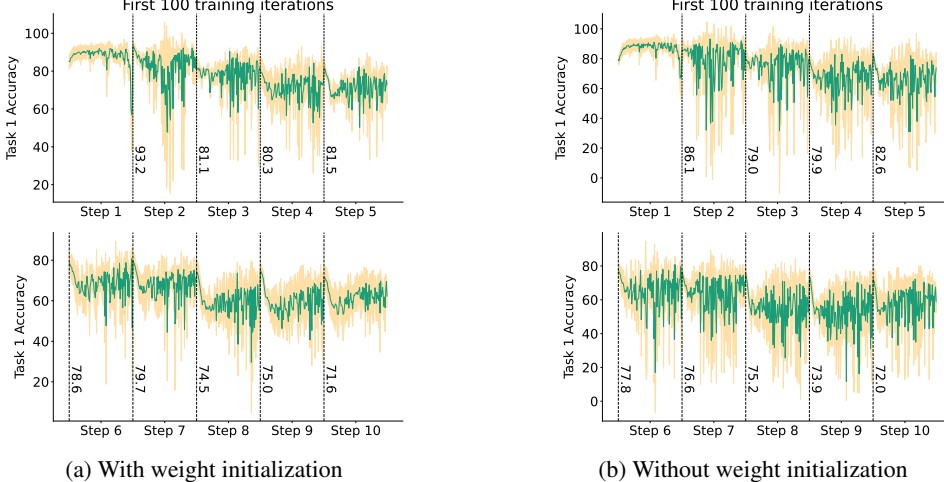

(a) With weight initialization          (b) Without weight initialization

Figure 12: **Effect of weight initialization on stability gap:** Test accuracy *with* and *without* weight initializations on the first task for the initial 100 iterations of incremental training on all ten tasks of CIFAR100. The green lines are the means over three different runs, the orange shades denote $\pm 1$ standard error of the mean. The labels to the vertical bars denote the accuracy values for the first iteration of training on each task.

the training iterations. To this end, Fig. 15 and Fig. 16 report the training iterations against the L2 norm of means and standard deviations for the past task heads (at each incremental training step) and the new task heads (at each training step). We observe two consistent trends regarding the evolution of distributions of the past and the new task heads. First, the proposed initialization of weights helps stabilize the learning of the means and standard deviations with (**red** against **green**) or without (**blue** against **orange**) regularizing the task distributions. Second, regularizing the task distributions increases the L2 norms of the learned mean and the standard deviation as these now have to encode more information to mimic the distributions of the hand-crafted text features.

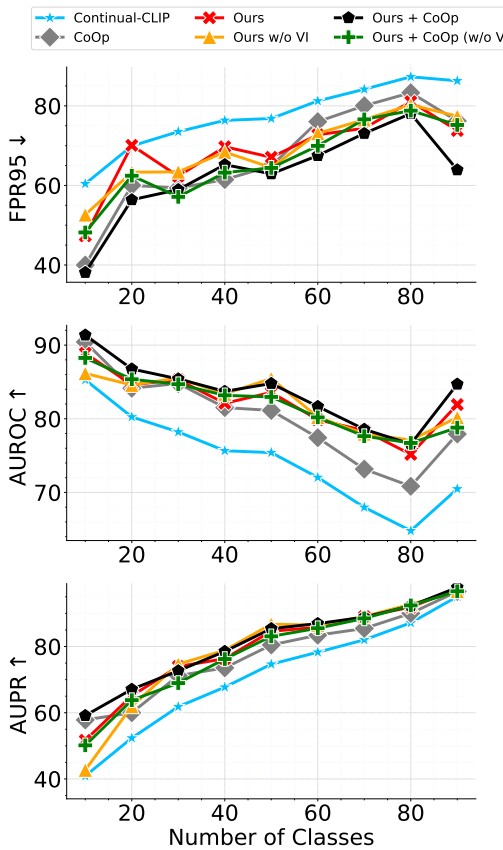

Figure 13: **Performance comparisons for post-hoc novel data detection** averaged over 3 runs on CIFAR100: FPR95 (**left**), AUROC (**middle**), and AUPR (**right**). The evaluations are carried over all but the last incremental test step.

# D Out-of-the-box utilities of probabilistic finetuning

## D.1 Post-hoc novel data detection

Our post-hoc novel data detection (PhNDD) setting aims to evaluate the continual learning methods at identifying novel data on the fly. To do so, we design an evaluation setup that uses no additional data resource other than that provided by the dataset-specific CL setting. Starting from the first test step, we treat the test data of the future tasks as *novel* while those of the seen tasks (including the most recently trained one) as *seen*. Since the last test step of a CL dataset has no future tasks, we exclude this step for our PhNDD evaluation, *i.e.,* we carry our PhNDD evaluation of CL models starting from the first until the penultimate test step.

Following other standard practices [75, 72], we use the Energy scores [75] of the outputs for each test sample as a measure of the model's confidence score. The samples assigned with a confidence score below the pre-defined confidence threshold are classified as novel. By assuming the seen data as the positive class and the novel data as the negative class, we can obtain a series of true positives rate (TPR) and false positive rate (FPR) by varying the confidence thresholds. One of our PhNDD evaluation metrics – the FPR95 then measures the FPR when the TPR is 0.95. As such, a lower FPR95 score indicates better PhNDD performance. Our other two PhNDD performance metrics include the the area under receiver operating characteristic curve (AUROC [73]) calculated based on FPR and TPR, and the precision-recall curve (AUPR [74]). Higher values of AUROC and AUPR indicate better PhNDD performance.

Table 4 reports the PhNDD metrics averaged over all the evaluated steps. Here, in Fig. 13, we show the evolution of these metrics with each evaluation starting from the first test step until the penultimate test step of CIFAR100. We observe that the zero-shot Continual-CLIP [14] has the poorest PhNDD performances (highest FPR95, and least AUROC and AUPR scores) across all steps given that it has not been finetuned on the downstream CL tasks. Among the finetuned methods, the CoOp [12] exhibits the poorest performances across all tasks. Among the variants of our method, combining CoOp with ours (CoOp + Ours) achieves the best PhNDD performances across all tasks. Furthermore, the deterministic versions: Ours w/o VI and CoOp + Ours (w/o VI) remain sub-optimal to their respective probabilistic variants, i.e., Ours and CoOp + Ours. The latter results validate the added perks of our probabilistic modeling framework for post-hoc novel data detection.

## D.2 Exemplar selection results

| Method | Avg | Last |
|---|---|---|
| CoOp | 76.71 | 64.1 |
| Clip-Adapter | 78.78 | 68.49 |
| Ours w/o VI | 84.44 | 76.55 |
| Ours | 85.18 | 77.92 |

Table 20: **Entropy-based** exemplar selection results for different methods on CIFAR100.

# E    Limitations and further research directions

Few potential directions of research for CLAP4CLIP include the design of: (a) parameter-efficient adapters [85] for very large CL settings; (b) better regularization techniques to alleviate forgetting; and (c) more informed [86] yet computationally efficient priors for inference. Similarly, along the direction of alleviating forgetting and mitigating the stability gap [57, 58], it would be interesting to see how class-specific prompts generated by pre-trained Large Language Models (LLMs) can be exploited to obtain task-relevant language-aware CLIP knowledge while preserving the zero-shot transfer ability of the learned prompts (see App. Table 21 for a preliminary investigation). Lastly, we consider applying CLAP4CLIP to more sophisticated Vision-Language tasks [87] as another possible direction for research. We elaborate further on each of these directions below.

**Parameter overhead.**    For each incoming task, CLAP4CLIP initializes a new head consisting of $d \times d$ parameters where $d$ is the output dimension of the CLIP model's encoder. For a very large number of real-world CL tasks, the number of finetunable parameters for CLAP4CLIP may thus become comparable to or larger than that of the pre-trained CLIP model's $\approx 150$ million parameters. For example, using a VIT-B/16 encoder with $d = 512$ brings an overhead of $\approx 525,000$ new parameters with each incoming task. After having seen $\approx 300$ new tasks, the number of CLAP parameters to be finetuned thus amount to $\approx 158$ million, which is larger than the frozen CLIP itself, and thus defeats the purpose of finetuning at the first place. One solid future direction to use CLAP4CLIP for very large real-world CL settings could thus be introducing more strict parameter-efficiency measures [88, 89] and/or learning probabilistic adapters with low-rank weights [85].

**Design choices.**    Other future directions for improving CLAP4CLIP could include the use of better regularization techniques to further prevent forgetting (see Table 13 for the current forgetting in CLAP), and the search for more informed yet computationally efficient priors (see Table 18 for the computational overhead attached with more informed priors).

**LLM-generated class descriptions as language-aware knowledge.**    In Sec. 3.3, we proposed using the text features from hand-crafted prompts as language-aware CLIP knowledge to help alleviate forgetting. However, hand-crafted prompts require manual labelling of data which is not always practical. Hence, several recent works [90, 91, 92] have opted to mining Large Language Models (LLMs) for efficiently obtaining the class-specific descriptions. To study the feasibility of alleviating forgetting using such LLM-generated class descriptions, we leverage the diverse prompts from CuPL [91] obtained using the GPT-3 [93] model. Our preliminary investigation suggests that the

hand-crafted prompts have an upper hand over GPT-3 based prompts for CLAP4CLIP performance (see Table 21). This could be because of the broad range of knowledge encoded in the GPT-generated prompts – which at times are irrelevant for the test images.

| Prompt type | Last ↑ | Avg ↑ | BwT ↑ | ECE ↓ | Runtime per iter. ↓ |
|---|---|---|---|---|---|
| Hand-crafted | **78.21** | **86.13** | -0.141 | **0.204** | 0.169 |
| GPT-3 | 77.76 | 85.7 | **-0.099** | 0.219 | **0.151** |

Table 21: **Performance comparison on CIFAR100** using hand-crafted vs. LLM-generated prompts for encoding language-aware CLIP knowledge. The results reported are averages over 3 runs. Best results across the metrics are highlighted in **bold**.

Based on the above finding, we suggest leveraging task-relevant LLM-generated descriptions as language-aware knowledge to be another promising future research direction. It is worth noting that a number of existing methods that rely on LLM-generated prompts are limited in their transferable knowledge across unseen classes and datasets [90, 91] (e.g., any new class at test-time would require mining the LLM descriptions in advance). On the contrary, our proposed weight initialization and task distribution regularization strategies provide a natural framework for LLM-generated prompts to be used alongside arbitrary learnable prompts (e.g. replacing $\mathbf{t}_y^{h,l}$ in eq. (7)). This compliments the idea of LLM-based text-only supervision frameworks [92] that seek to enrich *zero-shot transfer* of prompts to new classes by extracting rich contextual information from LLM data.[3]

**Compatibility with Vision-Language datasets**    The tasks we have covered so far in the paper are based solely on Vision datasets. To further demonstrate that our method is compatible with more sophisticated vision-language datasets, we here consider using a toy Visual Question Answering (VQAv2) task from the CLiMB dataset [87]. The CLiMB dataset hosts a number of tasks/settings to evaluate multi-modal and low-shot transfer abilities of CL algorithms. However, given the intricacies of these tasks (visual question answering, reasoning, etc.), we leave a full in-depth engagement with CLiMB [87] as a separate future direction for research.[4]

To show the aptness of our method for the dataset's tasks, we carry out preliminary experiments on the single-task learning setting [94] of the VQAv2 subset of CLiMB. Following [94], we rely on the BART model [95] for text generation here. Table 22 shows that our method surpasses the Continual-CLIP by 9.29 percentage points on the VQAv2 task, thus showing that ours enhances the zero-shot generalization capability of CLIP.

| Model | VQAv2 task score |
|---|---|
| Continual-CLIP | 57.42 |
| Ours | 66.71 |

Table 22: Single-task learning performance on the VQAv2 subset of the CLiMB dataset.

### E.1   Broader Impact

Recent years have witnessed the immense popularity of sequential generative models like Stable Diffusion [10] with applications in multimodal content generation as well as scientific research through fast and highly detailed sampling. The CLIP text encoder is widely employed by such generative models for learning personalized concepts conditioned on text prompts [96, 97]. By effective continual finetuning of the text encoder's features, our method can thus aid in customizing such models in a sequential manner using multiple, fine-grained concepts [98, 26].

---

[3]Given that new classes might emerge at test time for which we do not have the LLM-generated descriptions, it is important that the learned prompts preserve their zero-shot generalization ability.

[4]The CLiMB dataset [87] was introduced as an independent CL benchmark with a number of tasks (Visual Question Answering/Reasoning/Entailment) and training settings including low-shot and unimodal learning. Existing works [94] that study CLiMB thus rely solely on it *and not on additional datasets* for evaluations.

# F Derivation of ELBO for the static prior.

We seek to maximize the likelihood $p(y^{1:T})$ for all observed labels $y^{1:T}$. To derive the predictions, our framework uses the visual-aligned text features $\tilde{\mathbf{t}}_c^{1:T}$ and the image inputs $\mathbf{x}$ (see eq. (1)). Our evidence is thus $p(y^{1:T}|\mathbf{x};\tilde{\mathbf{t}}_c^{1:T})$ for which we derive the lower bound (ELBO). In the following, we denote the prior network as $p_\theta(z^t)$ for which the true posterior is $p_\theta(z^t|\mathbf{x};\tilde{\mathbf{t}}_c^t)$. We approximate the true posterior using the variational posterior $q_\phi(z^t|\mathbf{x};\tilde{\mathbf{t}}_c^t)$. Our derivation ends up with the reconstruction term $p_\theta(y^t|z^t,\mathbf{x};\tilde{\mathbf{t}}_c^t)$ that can be seen as a deterministic function converting a given latent vector $z^t$ and an input image $\mathbf{x}$ into an observation $y^t$. For our CLIP-based variational framework, this deterministic function is the cosine similarity operation followed by the softmax application (Eq. (3b)).

$$
\begin{aligned}
&\log p_\theta(y^{1:T}|\mathbf{x};\tilde{\mathbf{t}}_c^{1:T}) &&\text{(Log-likelihood of evidence)}\\
&= \log p_\theta(y^{1:T}|\mathbf{x};\tilde{\mathbf{t}}_c^{1:T})\int q_\phi(z^{1:T}|\mathbf{x};\tilde{\mathbf{t}}_c^{1:T})dz^{1:T} &&\left(\because \int q_\phi(z^{1:T}|x^{1:T})dz^{1:T}=1\right)\\
&= \int q_\phi(z^{1:T}|\mathbf{x};\tilde{\mathbf{t}}_c^{1:T})\left(\log p_\theta(y^{1:T}|\mathbf{x};\tilde{\mathbf{t}}_c^{1:T})\right)dz^{1:T} &&\text{(Bring evidence into integral)}\\
&= \mathbb{E}_{q_\phi(z^{1:T}|\mathbf{x};\tilde{\mathbf{t}}_c^{1:T})}\left[\log p_\theta(y^{1:T}|\mathbf{x};\tilde{\mathbf{t}}_c^{1:T})\right] &&\text{(Definition of Expectation)}\\
&= \sum_{t=1}^{T}\left[\mathbb{E}_{q_\phi(z^t|\mathbf{x};\tilde{\mathbf{t}}_c^t)}\left[\log p_\theta(y^t|\mathbf{x};\tilde{\mathbf{t}}_c^t)\right]\right] &&\text{(Rewrite using sum)}\\
&= \sum_{t=1}^{T}\left[\mathbb{E}_{q_\phi(z^t|\mathbf{x};\tilde{\mathbf{t}}_c^t)}\left[\log\frac{p_\theta(y^t,z^t|\mathbf{x};\tilde{\mathbf{t}}_c^t)}{p_\theta(z^t|\mathbf{x};\tilde{\mathbf{t}}_c^t)}\right]\right] &&\text{(Re-introduce $z^t$ by Chain rule of probability)}\\
&= \sum_{t=1}^{T}\left[\mathbb{E}_{q_\phi(z^t|\mathbf{x};\tilde{\mathbf{t}}_c^t)}\left[\log\frac{p_\theta(y^t,z^t|\mathbf{x};\tilde{\mathbf{t}}_c^t)q_\phi(z^t|\mathbf{x};\tilde{\mathbf{t}}_c^t)}{p_\theta(z^t|\mathbf{x};\tilde{\mathbf{t}}_c^t)q_\phi(z^t|\mathbf{x};\tilde{\mathbf{t}}_c^t)}\right]\right] &&\left(\text{Multiply by }1=\frac{q_\phi(z^t|\mathbf{x};\tilde{\mathbf{t}}_c^t)}{q_\phi(z^t|\mathbf{x};\tilde{\mathbf{t}}_c^t)}\right)\\
&= \sum_{t=1}^{T}\left[\mathbb{E}_{q_\phi(z^t|\mathbf{x};\tilde{\mathbf{t}}_c^t)}\left[\log\frac{p_\theta(y^t,z^t|\mathbf{x};\tilde{\mathbf{t}}_c^t)}{q_\phi(z^t|\mathbf{x};\tilde{\mathbf{t}}_c^t)}\right]+\mathbb{E}_{q_\phi(z^t|\mathbf{x};\tilde{\mathbf{t}}_c^t)}\left[\log\frac{q_\phi(z^t|\mathbf{x};\tilde{\mathbf{t}}_c^t)}{p_\theta(z^t|\mathbf{x};\tilde{\mathbf{t}}_c^t)}\right]\right] &&\text{(Split the expectation)}\\
&= \sum_{t=1}^{T}\left[\mathbb{E}_{q_\phi(z^t|\mathbf{x};\tilde{\mathbf{t}}_c^t)}\left[\log\frac{p_\theta(y^t,z^t|\mathbf{x};\tilde{\mathbf{t}}_c^t)}{q_\phi(z^t|\mathbf{x};\tilde{\mathbf{t}}_c^t)}\right]+\mathbb{D}_{\text{KL}}\left(q_\phi(z^t|\mathbf{x};\tilde{\mathbf{t}}_c^t)\|p_\theta(z^t|\mathbf{x};\tilde{\mathbf{t}}_c^t)\right)\right] &&\text{(Definition of KL divergence)}\\
&\geq \sum_{t=1}^{T}\left[\mathbb{E}_{q_\phi(z^t|\mathbf{x};\tilde{\mathbf{t}}_c^t)}\left[\log\frac{p_\theta(y^t,z^t|\mathbf{x};\tilde{\mathbf{t}}_c^t)}{q_\phi(z^t|\mathbf{x};\tilde{\mathbf{t}}_c^t)}\right]\right] &&(\because\text{ KL divergence}\geq 0)\\
&\geq \sum_{t=1}^{T}\left[\mathbb{E}_{q_\phi(z^t|\mathbf{x};\tilde{\mathbf{t}}_c^t)}\left[\log\frac{p_\theta(y^t|z^t,\mathbf{x};\tilde{\mathbf{t}}_c^t)p_\chi(z^t)}{q_\phi(z^t|\mathbf{x};\tilde{\mathbf{t}}_c^t)}\right]\right] &&\text{(Chain rule of probability)}\\
&\geq \sum_{t=1}^{T}\left[\mathbb{E}_{q_\phi(z^t|\mathbf{x};\tilde{\mathbf{t}}_c^t)}\left[\log p_\theta(y^t|z^t,\mathbf{x};\tilde{\mathbf{t}}_c^t)\right]+\mathbb{E}_{q_\phi(z^t|\mathbf{x};\tilde{\mathbf{t}}_c^t)}\left[\frac{p_\chi(z^t)}{q_\phi(z^t|\mathbf{x};\tilde{\mathbf{t}}_c^t)}\right]\right] &&\text{(Split the Expectation)}\\
&\geq \sum_{t=1}^{T}\left[\mathbb{E}_{q_\phi(z^t|\mathbf{x};\tilde{\mathbf{t}}_c^t)}\left[\log p_\theta(y^t|z^t,\mathbf{x};\tilde{\mathbf{t}}_c^t)\right]-\mathbb{D}_{\text{KL}}\left(q_\phi(z^t|\mathbf{x};\tilde{\mathbf{t}}_c^t)\|p_\chi(z^t)\right)\right] &&\text{(Definition of KL divergence)}
\end{aligned}
$$

# G Data-driven prior

The choice of prior is pivotal to a Bayesian inference workflow like ours [99]. While a standard Gaussian prior $p_\chi = \mathcal{N}(0, I)$ adapts well to a range of settings, it is (seemingly) uninformative regarding the nature of a given task [100]. With the end goal of deriving more informative priors, we thus seek to replace $p_\chi$ with task data-dependent prior distributions $p^t$, wherever applicable.

To this end, we first note that the outputs of the VGA module remain **invariant** not only to the order of the input text features (due to self-attention) but also to the order of the contextual image features (due to cross-attention). The latter invariance implies that the joint task-specific distribution learned by the encoder $q_\phi^t$ (conditioned on the VGA outputs $\hat{\mathbf{t}}_c^t$ from eq. 5a) is preserved if we were to permute the elements of the task-specific visual context set. More formally, this observation helps guarantee the (finite) *exchangeability* and the *consistency* properties of a stochastic process [101].

Motivated by the above, we treat the $t-$th task image features $\mathbf{x}^t$ as the target set $\mathcal{T}^t$ and employ a randomly chosen subset of it as our context set $\mathcal{C}^t$ to align the $t-$th task text features and to condition our prior $p^t$ on:

$$
\begin{aligned}
\hat{\mathbf{t}}_c^t &= \text{VGA}\left(Q=\mathbf{t}_c^t, K=V=\mathcal{C}^t\right),\\
p^t &= q_\phi^t(\tilde{\mathbf{t}}_c^t) = \left(\mu^t(\tilde{\mathbf{t}}_c^t),\sigma^t(\tilde{\mathbf{t}}_c^t)\right)
\end{aligned}
\tag{12}
$$

where $\tilde{\mathbf{t}}^t$ is the fused task-specific text feature following eq. (5b). The task-specific prior $p^t$ thus endows our training framework with a resemblance to the neural process (NP) architectures

[102, 103, 8]. Following NPs, we use the same encoder $q_\phi^t$ to parameterize the conditional prior and the variational posterior. This results in the following approximate ELBO (see App. F for the ELBO derivation):

$$
\begin{aligned}
\log p(\mathbf{y}^{1:T}|\mathbf{x}, \mathcal{C}^{1:T}) \geq & \\
\sum_{t=1}^{T} \Bigg[ \mathbb{E}_{q_\phi(z^t|\mathbf{x})} & \Big[ \log p(\mathbf{y}^t | z^t, \mathbf{x}_\mathcal{T}^t, \mathcal{C}^t) \Big] \\
& - \mathbb{D}_{\mathrm{KL}}\big(q_\phi(z^t|\mathcal{T}^t) \| q_\phi(z^t|\mathcal{C}^t)\big) \Bigg]
\end{aligned}
\tag{13}
$$

where in practice, the entire set of $t-$th images in a training minibatch form the target set $\mathcal{T}^t$ and a randomly chosen subset of the targets make up the context $\mathcal{C}^t$ [104]. Note that *unlike* NPs, our framework does not entirely rely on data-driven priors. Namely, while training on a CL task $t$, the past-task encoders are frozen and we have ample $t-$th task data points to condition the prior on. We thus resort to optimizing the ELBO (13) during training. On the other hand, during finetuning, we have limited task-specific data points to condition our context on. As such, we empirically found that switching to the static prior yields better results and thus resort to optimizing the ELBO (10) during finetuning.

### G.1 Effect of the context size on data-driven prior.

Table 18 in the main paper compares the results of using a data-driven prior against the uniform normal prior and the language-aware prior (see Sec. 3.3), where the latter is driven from the pre-trained text encoder using hand-crafted prompts. We observe that data-driven prior leads to minor accuracy improvements over the standard normal prior but falls narrowly behind the language-aware prior. Here, we study the influence of the batch size of the context set selected at random to derive our prior from.

Table 23 shows the last task accuracy with varying context sizes and a fixed target batch size of 64. We find that a smaller context set size hurts the performance of the model to the extent of falling behind the standard normal prior. Given that the context sets are the sampled subsets of the training (target) minibatches, a much smaller context set can lead to the increase in the prior matching loss values. We find that the context set batch size of 40 performs the best, and thus use this to ablate the prior-dependent performances in the main paper.

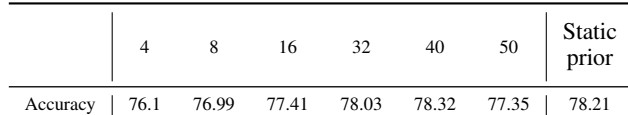

|  | 4 | 8 | 16 | 32 | 40 | 50 | Static prior |
|---|---|---|---|---|---|---|---|
| Accuracy | 76.1 | 76.99 | 77.41 | 78.03 | 78.32 | 77.35 | 78.21 |

Table 23: **Influence of the context set size** used to derive the data-driven prior on CIFAR100.

### G.2 Derivation of ELBO for the data-driven prior.

Similar to App. F, we start with the log-likelihood of the evidence which now involves conditioning on an additional context set $\mathcal{C}^{1:T}$. The $t-$th task context set is used to condition our prior network $p_\theta(z^t|\mathcal{C}^t)$. Following the standard practices of other data-driven prior frameworks [102, 8], we introduce parameter-sharing between our conditional prior and variational posterior networks. This allows us to replace our prior network with the variational posterior network $q_\phi(z^t|\mathcal{T}^t)$, where $\mathcal{T}$ is

the target set for task $t$.

$$\log p_\theta(Y_\mathcal{T}^{1:T}|\mathbf{x}_\mathcal{T}^{1:T}, \mathcal{C}^{1:T}) \qquad \text{(Log-likelihood of evidence)}$$

$$= \log p_\theta(Y_\mathcal{T}^{1:T}|\mathbf{x}_\mathcal{T}^{1:T}, \mathcal{C}^{1:T}) \int q_\phi(z^{1:T}|\mathbf{x}_\mathcal{T}^{1:T}, \mathcal{C}^{1:T})dz^{1:T} \qquad \left(\because \int q_\phi(z^{1:T}|\mathbf{x}_\mathcal{T}^{1:T}, \mathcal{C}^{1:T})dz^{1:T} = 1\right)$$

$$= \int q_\phi(z^{1:T}|\mathbf{x}_\mathcal{T}^{1:T}, \mathcal{C}^{1:T})\left(\log p_\theta(Y_\mathcal{T}^{1:T}|\mathbf{x}_\mathcal{T}^{1:T}, \mathcal{C}^{1:T})\right)dz^{1:T} \qquad \text{(Bring evidence into integral)}$$

$$= \mathbb{E}_{q_\phi(z^{1:T}|\mathcal{T}^{1:T})}[\log p_\theta(Y_\mathcal{T}^{1:T}|\mathbf{x}_\mathcal{T}^{1:T}, \mathcal{C}^{1:T})] \qquad \text{(By definition)}$$

$$= \sum_{t=1}^{T}\left[\mathbb{E}_{q_\phi(z^t|\mathcal{T}^t)}[\log p_\theta(Y_\mathcal{T}^t|\mathbf{x}_\mathcal{T}^t, \mathcal{C}^t)]\right] \qquad \text{(Rewrite using sum)}$$

$$= \sum_{t=1}^{T}\left[\mathbb{E}_{q_\phi(z^t|\mathcal{T}^t)}\left[\log \frac{p_\theta(Y_\mathcal{T}^t, z^t|\mathbf{x}_\mathcal{T}^t, \mathcal{C}^t)}{p_\theta(z^t|\mathbf{x}_\mathcal{T}^t, Y_\mathcal{T}^t, \mathcal{C}^t)}\right]\right] \qquad \text{(Re-introduce } z^t \text{ by Chain rule of probability)}$$

$$= \sum_{t=1}^{T}\left[\mathbb{E}_{q_\phi(z^t|\mathcal{T}^t)}\left[\log \frac{p_\theta(Y_\mathcal{T}^t|z^t, \mathbf{x}_\mathcal{T}^t, \mathcal{C}^t)\, p_\theta(z^t|\mathcal{C}^t)}{p_\theta(z^t|\mathcal{T}^t)}\right]\right] \qquad \text{(By Chain rule of probability; } \mathcal{C} \subset \mathcal{T})$$

$$= \sum_{t=1}^{T}\left[\mathbb{E}_{q_\phi(z^t|\mathcal{T}^t)}\left[\log \frac{p_\theta(Y_\mathcal{T}^t|z^t, \mathbf{x}_\mathcal{T}^t, \mathcal{C}^t)\, p_\theta(z^t|\mathcal{C}^t)\, q_\phi(z^t|\mathcal{T}^t)}{p_\theta(z^t|\mathcal{T}^t)\, q_\phi(z^t|\mathcal{T}^t)}\right]\right] \qquad \text{(Equivalent fraction)}$$

$$= \sum_{t=1}^{T}\left[\mathbb{E}_{q_\phi(z^t|\mathcal{T}^t)}\left[\log p_\theta(Y_\mathcal{T}^t|z^t, \mathbf{x}_\mathcal{T}^t, \mathcal{C}^t)\right]\right.$$
$$\left. + \mathbb{E}_{q_\phi(z^t|\mathcal{T}^t)}\left[\log \frac{p_\theta(z^t|\mathcal{C}^t)}{q_\phi(z^t|\mathcal{T}^t)}\right] + \mathbb{E}_{q_\phi(z^t|\mathcal{T}^t)}\left[\log \frac{q_\phi(z^t|\mathcal{T}^t)}{p_\theta(z^t|\mathcal{T}^t)}\right]\right] \qquad \text{(Split the expectation)}$$

$$= \sum_{t=1}^{T}\left[\mathbb{E}_{q_\phi(z^t|\mathcal{T}^t)}\left[\log p_\theta(Y_\mathcal{T}^t|z^t, \mathbf{x}_\mathcal{T}^t, \mathcal{C}^t)\right]\right.$$
$$\left. - \mathbb{D}_{\text{KL}}\left(q_\phi(z^t|\mathcal{T}^t)\|p_\theta(z^t|\mathcal{C}^t)\right) + \mathbb{D}_{\text{KL}}\left(q_\phi(z^t|\mathcal{T}^t)\|p_\theta(z^t|\mathcal{T}^t)\right)\right] \qquad \text{(By definition of KL divergence)}$$

$$\geq \sum_{t=1}^{T}\left[\mathbb{E}_{q_\phi(z^t|\mathcal{T}^t)}\left[\log p_\theta(Y_\mathcal{T}^t|z^t, \mathbf{x}_\mathcal{T}^t, \mathcal{C}^t)\right] - \mathbb{D}_{\text{KL}}\left(q_\phi(z^t|\mathcal{T}^t)\|p_\theta(z^t|\mathcal{C}^t)\right)\right] \qquad (\because \text{KL divergence} \geq 0)$$

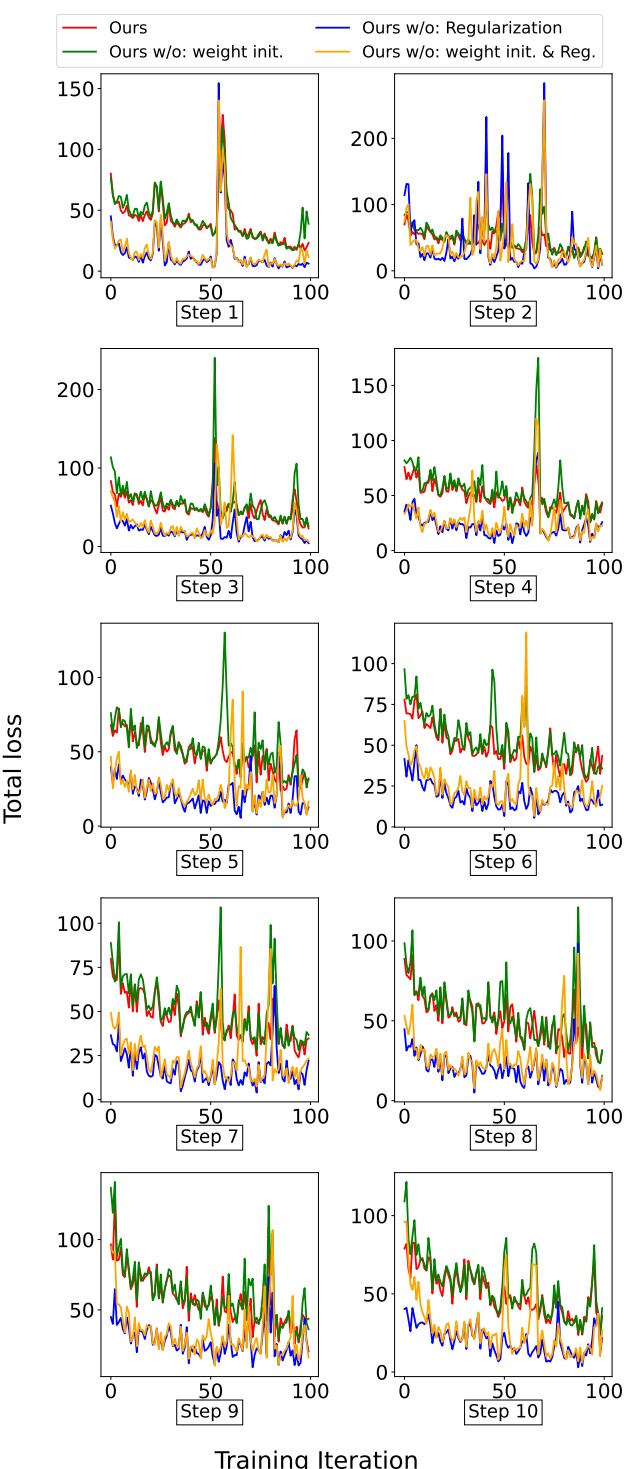

Figure 14: **Evolution of the loss value** $\mathcal{L}$ during the first 100 training iterations of each task on CIFAR100. Training with our proposed weight initialization strategy consistently leads to lower training losses thus bridging the **stability gap** [57] in CL.

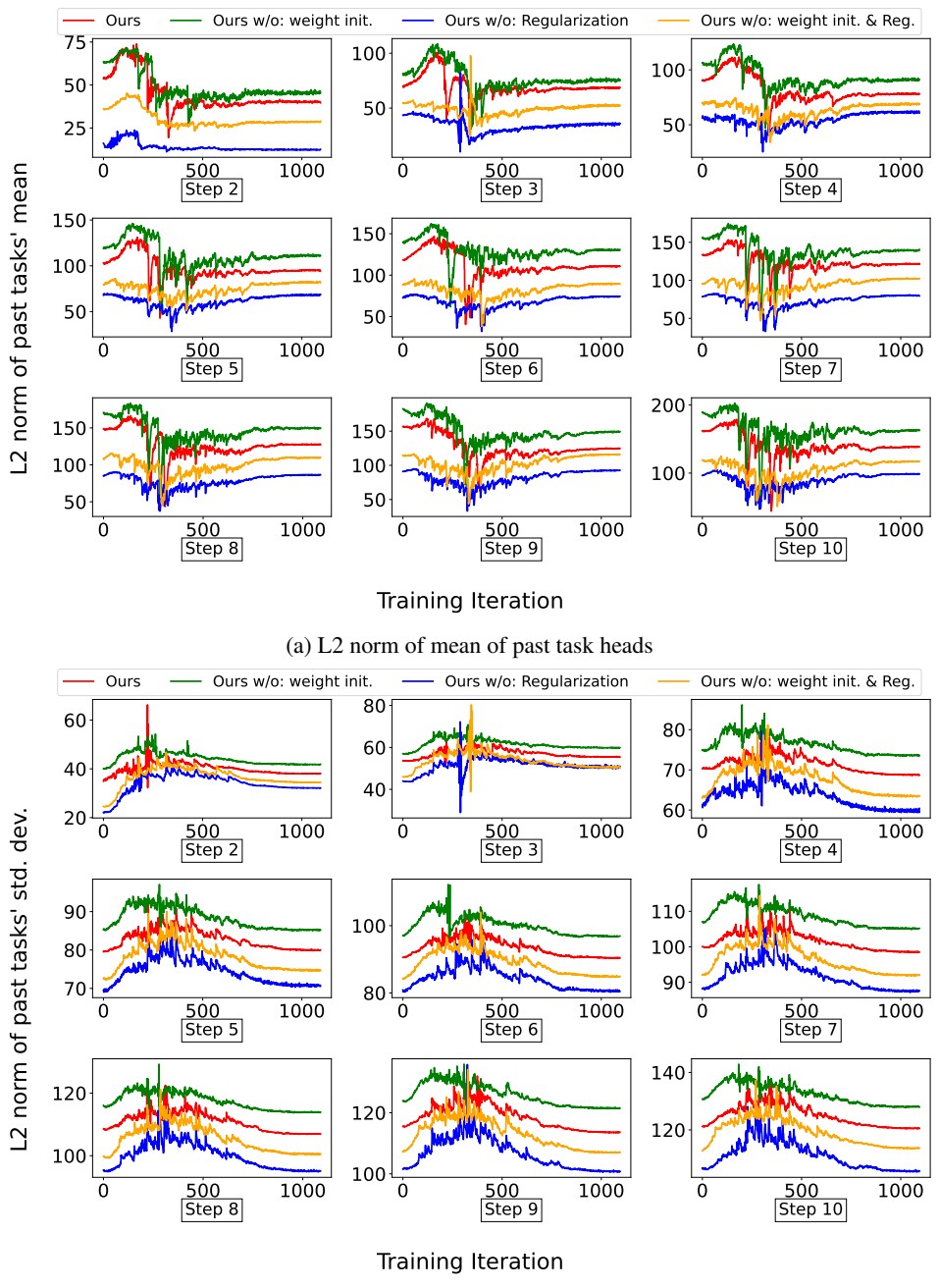

(a) L2 norm of mean of past task heads

(b) L2 norm of standard deviation of past task heads

Figure 15: **Evolution of mean and standard deviation** of past task encoders with training iterations.

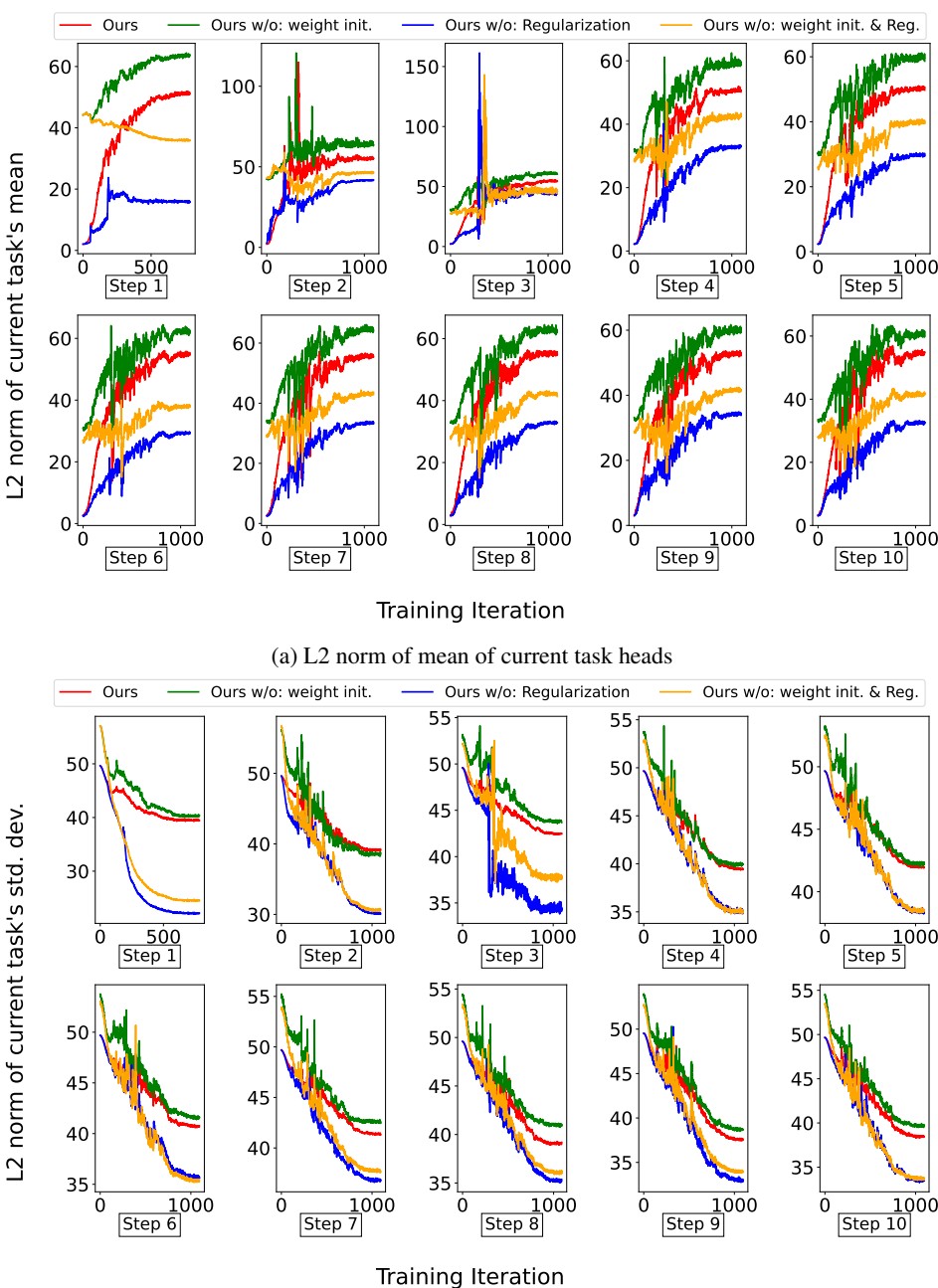

(a) L2 norm of mean of current task heads

(b) L2 norm of standard deviation of current task heads

Figure 16: **Evolution of mean and standard deviation** of task encoders (recorded at the step where they were first introduced) with training iterations.

