# OpenReview forum: "CLAP4CLIP: Continual Learning with Probabilistic Finetuning for Vision-Language Models"
_NeurIPS.cc/2024/Conference — NeurIPS 2024 poster_

### Official Review · Reviewer_ey1B · 2024-07-09

**Soundness:** 4
**Presentation:** 4
**Contribution:** 4
**Rating:** 8
**Confidence:** 3

**Summary:**

The authors present a novel method for CL called CLAP (Continual LeArning with Probabilistic finetuning) applied to the CLIP model. The technique employs probabilistic modeling to refine task-specific modules aligned with visual-guided text features, improving model adaptation to new tasks while mitigating the forgetting of old ones. It also benefits from the knowledge of pre-trained CLIP for this purpose. This method also offers compatibility with prompt-based finetuning methods as well.  The approach demonstrates superiority over traditional deterministic finetuning methods through enhanced performance and better uncertainty estimation in diverse experimental setups.

**Strengths:**

1. **Originality and Significance:** The paper introduces a unique approach to CL by integrating probabilistic finetuning with the CLIP model. The proposed pipeline is innovative and significant, as it addresses the critical issue of catastrophic forgetting in CLIP when trained on streams of tasks using well-justified solutions.

2. **Quality and Clarity:** The paper is well-written, with clear explanations of the methodology and its advantages.

3. **Technical Soundness:** The experimental setup is comprehensive, covering several datasets and comparative baselines. The results convincingly demonstrate the effectiveness of CLAP4CLIP in improving in-domain performance and generalization to new tasks. Especially achieving positive back-ward transfer in VTAB (Table 12) was a very interesting finding.

**Weaknesses:**

1. **Presentation Issues:** Figure 1 could be larger to improve readability and clarity. The paper would benefit from better visualization to help convey the complex mechanisms of the proposed method more effectively.

2. **Reference Order and Citations:** The order of the references seems incorrect, which could potentially confuse readers. Additionally, Reference [36] should be discussed in the related works section as well to better contextualize the contributions of the paper.

**Questions:**

Please address the mentioned weaknesses.

**Limitations:**

The limitations of this work are discussed and presented in Appendix E.

---

> ### Author Rebuttal · Authors · 2024-08-06
>
> Dear Reviewer ey1B,
>
> Thank you for your comments.
>
> - We assure you that we will enlarge Figure 1 in the next version of the paper to improve its readability and clarity.
>
> - We will correct the reference order accordingly and discuss the relevant reference in the related works. We apologize for any confusion this may have created.

---

> > ### Comment · Reviewer_ey1B · 2024-08-11
> >
> > Thank you to the authors for addressing my concerns. I will be keeping my original score.

---

### Official Review · Reviewer_KrxH · 2024-07-12

**Soundness:** 2
**Presentation:** 3
**Contribution:** 2
**Rating:** 4
**Confidence:** 4

**Summary:**

This paper introduces CLAP4CLIP, a method designed to enhance continual learning (CL) using CLIP, a pre-trained vision-language model. The method leverages probabilistic fine-tuning with task-specific adapters to mitigate the issue of catastrophic forgetting commonly faced in CL. By incorporating visual-guided attention (VGA) modules, the model aims to align text features with frozen visual features during incremental training. The proposed method is evaluated on several datasets, including CIFAR100, ImageNet100, ImageNet-R, CUB200, and VTAB, and compared against multiple baselines and state-of-the-art fine-tuning methods.

**Strengths:**

1. The paper presents a novel approach by integrating probabilistic fine-tuning and visual-guided attention into CLIP for continual learning. The use of task-specific adapters and the Bayesian variational inference framework adds a unique angle to the existing methods.
2. The methodology is well-structured, and the experiments are thorough, comparing the proposed method against a wide range of baselines and state-of-the-art approaches.
3. Addressing catastrophic forgetting in CL is a significant challenge, and the proposed method offers a promising solution. The integration with CLIP, known for its zero-shot learning capabilities, highlights the potential impact on real-world applications.

**Weaknesses:**

1. Given CLIP’s strong generalization abilities and its effectiveness in zero-shot learning, the reliance on replay strategies might seem redundant and potentially underutilizes CLIP’s full capabilities.
2. The paper does not explicitly detail whether the VGA module is updated during incremental tasks. This is a critical aspect, as the update strategy could significantly impact the model's performance and stability.
3. The method involves multiple components, such as probabilistic fine-tuning, VGA modules, and task-specific adapters, which may introduce significant computational overhead.

**Questions:**

1. Could you provide more insights into why the replay strategy is necessary and how it complements the use of CLIP in your method?
2. How is the VGA module handled during incremental tasks? Are its parameters updated, and if so, what strategy is used to ensure consistency and stability across tasks?
3. What are the computational requirements of your method compared to the baselines, especially concerning the additional components introduced? How do you balance the trade-off between performance improvements and computational efficiency?

**Limitations:**

*

---

> ### Author Rebuttal · Authors · 2024-08-06
>
> Dear Reviewer KrxH,
>
> Thank you for your comments and suggestions. In what follows, we have tried our best to address your concerns.
>
> - **Our method stands out without replay, replay further boosts its performance:** Thank you for your comment. We would first like to mention that our additional rebuttal experiments without memory replay (Table 2 in the rebuttal pdf) clearly show that our method outperforms the compared SOTA even without replay.
>
> The use of memory replay in our proposed continual learning method is thus not redundant, but rather complementary to CLIP's existing strengths. CLIP's power primarily comes from its strong alignment between visual and textual modalities. As we finetune the model to incremental tasks, memory replay helps maintain this crucial alignment for previously learned tasks. Without it, the model might drift, compromising its image-text retrieval performance and in turn, its zero-shot capabilities on previously seen concepts while learning new ones. More importantly, like other CL setups, our use of replay is confined to the training process. Once the model is fine-tuned, there's no additional computational overhead during inference brought about by memory replay.
>
> In addition, from the perspective of prior regularization, memory replay has broadly been used for adapting/finetuning a range of foundation models, e.g. the adaptation of vision and language models [1], latent diffusion models [2], and large language models [3]. These practical scenarios often demand that the deployed foundation models maintain their zero-shot transfer ability as well as their inference-time efficiency. Replay thus helps boost the former without affecting the latter.
>
> Lastly, in Section 5 of the main paper, we discuss that our proposed method is **agnostic** of the strategy used to select exemplars for memory replay. To support this, in Table 18, Appendix D.2, we show that our method works well even with an entropy-based exemplar selection strategy – a setup where existing deterministic methods lag due to their poor predictive confidences.
>
> - **Working of VGA module:** Thank you for your comment and apologies for the working of the VGA module being unclear from the main paper – we mention it explicitly in Appendix A.2 “Training for memory consolidation”.  The VGA module is indeed shared across different tasks and its parameter is updated normally during the training phase when our training data comprises the current task data as well as the replay memory data. This is followed by the training phase for memory consolidation where we finetune our model on the class-balanced dataset of new data and rehearsal data. Following other well-established parameter-isolation CL algorithms [4-5], here we freeze the task-shared VGA parameters to avoid interference with the knowledge acquired during the normal training phase (given that during the consolidation phase, our training data is vastly reduced and comprises only the class-balanced data maintained in the small replay buffer).
>
> - **Computational requirements of our method:** We provide a detailed comparison of the parameter and time analyses for different methods in Fig. 4 and Appendix Table 16, respectively. A major computational overhead for our proposed probabilistic method is the number of Monte Carlo samples. In App C.1, we thoroughly report the accuracy-runtime tradeoff for our proposed method with the varying numbers of MC samples. Namely, the accuracy remains poorer in the range [1,10], grows in the range [10, 20], and in general, saturates thereafter. On the other hand, the runtime grows roughly by 103% as we increase the number of MC samples from 1 to 50.
>
> From the perspective of probabilistic finetuning, we also analyze the performance vs efficiency trade-off for the choice of prior type in Appendix C.4. As shown in Table 15, compared to the static standard normal prior, the data-driven and language-aware priors offer us minor performance gains in terms of last and avg. accuracy and backward transfer scores. However, these performance gains are neutralized by the higher cost of runtime per inference iteration. As a result, we stick to using the standard normal prior throughout our work.
>
> References:
>
> [1] Smith, James Seale *et al.* “Adaptive Memory Replay for Continual Learning.” CVPR 2024 Workshops.
>
> [2] Kumari, Nupur *et al.* “Multi-Concept Customization of Text-to-Image Diffusion.” CVPR 2022.
>
> [3] Wang, Yifan *et al.* “InsCL: A Data-efficient Continual Learning Paradigm for Fine-tuning Large Language Models with Instructions.” ACL 2024.
>
> [4] Castro, Francisco Manuel *et al.* “End-to-End Incremental Learning.” ECCV 2018.
>
> [5] Douillard, Arthur *et al.* “DyTox: Transformers for Continual Learning with DYnamic TOken eXpansion.” CVPR 2022.

---

> > ### Author Response · Authors · 2024-08-11
> > **Requesting feedback on our rebuttal**
> >
> > Dear Reviewer KrxH,
> >
> > We thank you again for taking the time to review this work. We put our best effort into preparing the rebuttal to your questions, including running experiments without memory replay. We would very much appreciate it if you could provide us with your feedback on our rebuttal. We would be glad to answer any further questions and clarify any concerns.
> >
> > Also, if you are satisfied with our answers, please consider revising your score.
> >
> > With best regards

---

### Official Review · Reviewer_bibK · 2024-07-13

**Soundness:** 2
**Presentation:** 3
**Contribution:** 3
**Rating:** 3
**Confidence:** 4

**Summary:**

The paper emphasizes on the existing limitations of deterministic approaches in fine-tuning and highlights the need for probabilistic fine-tuning approach. Following this, it proposes a probabilistic parameter efficient fine-tuning method for continually learning vision language models like CLIP.

**Strengths:**

1. The proposed approach seems novel.

**Weaknesses:**

1. Justification of probabilistic modelling of the text feature and not the image feature space is not clear.
2. The approach is highly inefficient in terms of inference time.
3. Recent approaches like ConvPrompt[a], CODA-Prompt[b], HiDe-Prompt[c], and SLCA[d] not compared.


[a]Roy, Anurag, et al. "Convolutional Prompting meets Language Models for Continual Learning." Proceedings of the IEEE/CVF Conference on Computer Vision and Pattern Recognition. 2024

[b]Smith, James Seale, et al. "Coda-prompt: Continual decomposed attention-based prompting for rehearsal-free continual learning." Proceedings of the IEEE/CVF Conference on Computer Vision and Pattern Recognition. 2023.

[c]Wang, Liyuan, et al. "Hierarchical decomposition of prompt-based continual learning: Rethinking obscured sub-optimality." Advances in Neural Information Processing Systems 36 (2024).

[d]Zhang, Gengwei, et al. "Slca: Slow learner with classifier alignment for continual learning on a pre-trained model." Proceedings of the IEEE/CVF International Conference on Computer Vision. 2023.

**Questions:**

1. Could you address the points I raised in the weakness section?

2. The paper demonstrates a ~2% performance improvement with the addition of the memory consolidation component. However, I'm interested in seeing how the proposed method performs when the replay memory size is set to zero.

**Limitations:**

The authors have discussed the limitations of their work in the paper.

---

> ### Author Rebuttal · Authors · 2024-08-06
>
> Dear Reviewer bibK,
>
> Thank you for your comments and suggestions. In what follows, we have tried our best to address your concerns.
>
> - **Why do we do probabilistic modeling of text feature space?** We opt for probabilistic modeling of task-specific text feature space rather than image feature space mainly in light of the practical constraints imposed by the class-incremental learning (CIL) setting. In CIL, at test time, we are not given the task labels for images. As such, if we were to use task-specific adapters to model task-specific visual features distribution (rather than task-specific text features distribution as we do now), then we must know which images should be routed to what adapter – something not plausible at test time. A naive getaway would be to route the text-guided visual features to all available adapters and then infer the correct prediction based on the adapter outputs. Such an exhaustive routing mechanism would greatly increase our computational burden at test time. Modeling the distribution of visual-guided text features helps us overcome this because now our visual features serve as a shared context to which all task-specific text features (which we can distinguish simply by their labels) can attend. By sampling from the distributions over such visual-guided task-specific text features, we can compute their cosine similarities with the visual features to obtain our predictive logits.
>
> - **Comparison with zero replay memory size:**  In Table 2 of the rebuttal pdf, we have provided a comparison of our proposed method with the SOTA models for continual learning without replay with ViT (i.e., CODA-Prompt) and with CLIP (i.e., AttriCLIP). Here, leveraging the instance-conditioned and semantically diverse prompts of AttriCLIP provides an edge. Our variant leveraging AttriCLIP further improves its performance surpassing the SOTA. In the bottom two rows of the table, we ablate the role of our proposed language-aware distribution regularization and weight init. components and find that the former is crucial in avoiding forgetting in this setting.
>
>
> - **Inference time overhead:** The inference time of our method is highly dependent on the number of MC samples as well as the prompt type being used. For instance, in Table 16 Appendix C.5, we show that using multi-modal prompts (MaPLe) with Ours leads to an average inference time of 0.064s which is comparable to and even lower than other existing methods like AttriCLIP (0.257s). Also, in our rebuttal to Reviewer **giBN**, we state that the inference time of several variants of our method remains lower than the compared SOTA (PROOF) -- 0.163s (Ours) vs 0.177s (PROOF).
>
> We nonetheless agree that MC sampling for probabilistic modeling endows our method with higher inference time overhead compared to other deterministic methods (something we have thoroughly ablated in Fig. 8, Appendix C.1). However, such a caveat is known to be general for probabilistic models, and we believe that rejecting our method based on this (while downweighing the wide range of performance advantages) would be unfair. Put together, our proposed method outperforms several existing SOTA with better accuracy, backward and forward (zero-shot) transfer ability, and calibration, and also performs better on settings without memory replay and with restricted computational budget, as shown in Table 2 and Table 4 in the rebuttal pdf, respectively.
>
> - **Comparison with every single recent work:** We would like to state that continual learning with pre-trained foundation models is a rapidly evolving field, as is the evolution of new pre-trained foundation models itself. Given this fast-moving environment, it is challenging to provide an exhaustive comparison with every single recent work. While we strive for comprehensive analysis, we hope the reviewer understands that an all-encompassing comparison may not be practically achievable within the scope of this work. That said, we have thoroughly covered the **most relevant** SOTA models for vision-language models (PROOF and AttriCLIP) as well as for vision-only models (DualPrompt and CODA-Prompt) across a range of datasets and settings.

---

> > ### Author Response · Authors · 2024-08-11
> > **Requesting feedback on our rebuttal**
> >
> > Dear Reviewer bibK,
> >
> > We thank you again for taking the time to review this work. We put our best effort into preparing the rebuttal to your questions, including reporting the experiments on zero replay memory size, comparing with CODA-Prompt, and justifying design choices/inference time overhead. We would very much appreciate if you could engage with us through your feedback on our rebuttal. We would be glad to answer any further questions and clarify any concerns.
> >
> > Also, if you are satisfied with our answers, please consider revising your score.
> >
> > With best regards

---

### Official Review · Reviewer_giBN · 2024-07-15

**Soundness:** 3
**Presentation:** 3
**Contribution:** 3
**Rating:** 5
**Confidence:** 4

**Summary:**

This paper proposes Continual Learning with Probabilistic Finetuning (CLAP) for class-incremental learning using CLIP. The key modules of the proposed idea are as follows. First, the authors introduce a CLIP-based probabilistic finetuning model using Bayesian Variational Inference to achieve better generalization during the continual learning process. Second, they propose a visual-guided attention (VGA) model to facilitate cross-modal feature alignment in the continual learning process. Lastly, to alleviate forgetting of pre-trained language-aware CLIP knowledge, they suggest past-task distribution generalization. Additionally, to enable parameter-efficient learning, a probabilistic adapter is used, along with a method for its initialization to enhance stability. Experimental results on various datasets demonstrate that the proposed algorithm achieves superior class-incremental learning performance compared to existing algorithms.

**Strengths:**

The strengths of this paper are as follows:

1. The paper is well-written and easy to understand.

2. The proposed modules for successful class-incremental learning using CLIP are meticulously designed. Through various analyses and ablation studies, the roles of each module are thoroughly demonstrated. Although the algorithm appears somewhat complex compared to existing algorithms, parameter and time analyses show that the actual cost difference is negligible.

3. In class-incremental learning experiments using diverse datasets, the proposed algorithm consistently outperforms existing algorithms across various evaluations.

**Weaknesses:**

1. I have no major concerns regarding the contribution of the proposed algorithm for class-incremental learning using CLIP. However, I have some questions based on the experimental results.

1-1) Unlike algorithms like L2P and DualPrompt, which utilize exemplar memory, the paper does not use exemplar memory. Are the results in L2P and DualPrompt sections of the paper based on using exemplar memory in a fair comparison? To ensure a fair comparison, results using exemplar memory should be shown.

1-2) The paper only considers prompt-based algorithms like L2P and DualPrompt using Vision Transformer (ViT). However, newer prompt-based algorithms (e.g., CODA-Prompt) that achieve better performance and representation-based algorithms (e.g., Ranpac and EASE) that generally improve performance have been proposed. The authors should consider these algorithms as additional baselines. For more details, please refer to this survey paper [1].

1-3) Among existing baselines, PROOF is currently the state-of-the-art algorithm. While Table 1 shows PROOF's results, Tables 2 and 16 do not. To validate the superiority of the proposed algorithm, results for PROOF should be shown in these tables and other key experiments.

2. Recent research has highlighted discussions on hyperparameter tuning for class-incremental learning algorithms [2] and pointed out issues regarding computational costs[3]. From this perspective, I believe that further discussion and additional analysis on computational costs regarding the proposed algorithm would make the paper more convincing.

3. Lastly, I have personal concerns regarding the necessity of class-incremental learning research using CLIP. As shown in [1], algorithms using only ViT have already achieved excellent performance in class-incremental learning research. Despite using pretrained ViT, one of their major advantages is achieving superior performance without using exemplar memory. Considering this perspective, the proposed research not only uses a pretrained CLIP model but also requires text information and additional exemplar memory usage. Simply comparing numerical experimental results suggests that the proposed algorithm does not achieve overwhelmingly superior performance compared to algorithms using only pretrained ViT. In light of this, I would like to understand: 1) What justifies the necessity of class-incremental learning research using CLIP with exemplar memory? Additionally, 2) I am curious about the potential for the proposed algorithm's ideas to be applied to continual learning in other domains using CLIP or continual pre-training of CLIP itself.

[1] "Continual learning with pre-trained models: A survey." arXiv preprint arXiv:2401.16386 (2024).

[2] "Hyperparameters in Continual Learning: a Reality Check." arXiv preprint arXiv:2403.09066 (2024).

[3] "Computationally budgeted continual learning: What does matter?." Proceedings of the IEEE/CVF Conference on Computer Vision and Pattern Recognition. 2023.

**Questions:**

See the Weakness section. I don't have significant concerns about the algorithm proposed in this paper, but I couldn't give it a higher score due to some experimental uncertainties and questions about the necessity of the setting. If the authors address these concerns in the rebuttal, I would gladly raise my score.

**Limitations:**

There is no potential negative societal impact of this paper.

---

> ### Author Rebuttal · Authors · 2024-08-06
>
> Dear Reviewer giBN,
>
> Thank you for your comments and suggestions. We have tried our best to address your concerns below.
>
> 1-1) **Fair comparison with L2P and DualPrompt:**
> The scores we report for L2P and DualPrompt are fair and use a CLIP-based backbone as well as memory replay with the same number of exemplars as our method. As also mentioned in our global rebuttal, both L2P and DualPrompt originally use the ViT-B/16 backbone pre-trained on the ImageNet-21K dataset [1] which leads to higher performance scores on the compared datasets. However, replacing this pretrained ViT backbone with that of OpenAI CLIP’s pretrained ViT backbone leads to a drop in the performance across all datasets and clearly necessitates the need for memory replay to catchup with our proposed method. The aforesaid drop in performance has also been observed by other studies adapting L2P and DualPrompt for continual learning with CLIP [2].
>
> 1-2) **CODA-Prompt as baseline:** Based on your comment, we have included CODA-Prompt (reimplemented using the pre-trained OpenAI CLIP’s ViT backbone) into our baselines. Like L2P and DualPrompt, the original CODA-Propmt paper uses ImageNet-21K pre-trained ViT. Replacing this with CLIP’s ViT backbone necessitates the need for memory replay (see Table 1 vs Table 2 in the rebuttal pdf) to catch up with the performance of our method. Hence, our comparison with CODA-P, L2P, and DualPrompt remains fair and consistent.
>
> 1-3) Thank you for your concern. We have included the comparison with PROOF on the Cross-Dataset Continual Learning setup in Table 3 of the rebuttal pdf.
>
> Also, adding to Table 16, the avg. inference time for PROOF remains 0.177s which is slower than the base (Ours) and the MaPLe variant of ours but faster than our other variants utilizing task-conditioned (CoOp) and instance-conditioned (AttriCLIP) prompts.
>
> 2) **Computationally-budged CL setup:** Regarding computationally-budgeted CL setup, we would like to state that not all CL training setups require to be computationally-budgeted. In fact, upon finetuning of large pre-trained models like CLIP, it is often the case that their performance post-deployment, both in terms of accuracy/zero-shot transfer ability and inference time, counts the most. Our thorough evaluations make a clear statement about the upper hand of our method in terms of performance based on a number of metrics – accuracy, backward and forward (zero-shot) transfer ability, and calibration. Our inference time is also mainly dependent on the type of prompts used with our method.
>
> In Table 4 of the rebuttal pdf, we have nevertheless provided an ablation on the “normal” budgeted CL setup of [3] where, on each incremental training task, we allocate the number of training iterations equivalent to 1 epoch on the first (base) task of each dataset. Here, our variant utilizing instance-conditioned prompts of AttriCLIP outperforms other compared methods. A further ablation shows that our proposed weight distribution regularization technique indeed remains a crucial component at tackling forgetting on the budgeted setup (see the two bottom-most rows in Table 4).
>
> 3) **“The proposed algorithm does not achieve overwhelmingly superior performance .. “:** In light of the consistent superior performance scores of our method without using additional exemplar memory replay and on computationally budgeted CL setup, we hope that the reviewer reconsiders their remark.
>
> 4) **Justification on exemplar memory:** We would like to state that the use of memory replay in our proposed continual learning method is not redundant, but rather complementary to CLIP's existing strengths. To justify this, in Table 2 of the attached pdf, we show that our method is indeed compatible with setups without memory replay, and performs either on par or better than other compared methods, including ViT-only methods.
>
> Moreover, we would like to highlight two such perspectives to clarify that the usage of a small additional exemplar memory is not to be seen as an overhead in continual learning. **First,** for pre-trained foundation models, a critical performance measure for deploying their  (incrementally) finetuned variants is often their inference time overhead and their downstream/zero-shot transfer abilities. The use of exemplar memory does not affect the inference time overhead of these methods. In fact, comparing Tables 1 and 2 from the rebuttal pdf, it is clear that using replay memory boosts their downstream task performance. Lastly, we clearly state the superior zero-shot (forward) and backward transfer abilities of our proposed method in Sec. 4.1 in the main paper. **Second,** if we were to look at exemplar memory from a prior regularization perspective, then it becomes immediately clear how predominant the usage of these is across different domains, e.g. the adaptation of Vision-Language Models [4], latent diffusion models [5], and LLMs [6].
>
> References:
>
> [1] Dosovitskiy, Alexey *et al.* “An Image is Worth 16x16 Words: Transformers for Image Recognition at Scale.” ICLR 2021.
>
> [2] Zhou, Da-Wei *et al.* “Learning without Forgetting for Vision-Language Models.”
>
> [3] Prabhu, Ameya *et al.* “From Categories to Classifier: Name-Only Continual Learning by Exploring the Web.” ArXiv abs/2311.11293 (2023).
>
> [4] Smith, James Seale *et al.* “Adaptive Memory Replay for Continual Learning.” CVPR 2024 Workshops.
>
> [5] Kumari, Nupur *et al.* “Multi-Concept Customization of Text-to-Image Diffusion.” CVPR 2022.
>
> [6] Wang, Yifan *et al.* “InsCL: A Data-efficient Continual Learning Paradigm for Fine-tuning Large Language Models with Instructions.” ACL 2024.

---

> > ### Comment · Reviewer_giBN · 2024-08-09
> >
> > I would like to thank the authors for conducting additional experiments and providing an author response to address the concerns raised in my review. I have thoroughly read both the author response and the updated PDF, and as a result, I am increasing my score to 5. I hope that the final version of the paper will fully incorporate the contents of the response.

---

### Author Rebuttal · Authors · 2024-08-06

We would like to thank the reviewers for their comments and constructive suggestions on our manuscript. Here, we provide a pdf containing results for the experiments asked in the reviews. We also highlight the three major points raised in the reviews and how our rebuttal has addressed these:

- **Comparison with CODA-Prompt:** We have included CODA-P into our baselines and have compared it with our method across all datasets with and without replay in Table 1 and 2 of the rebuttal pdf, respectively. Note that similar to L2P and DualPrompt, CODA-P paper uses ImageNet-21K pre-trained ViT-B/16 model as the baseline. Given the similarity of ImageNet-21K images with the compared datasets, this leads to quite high performances.  For a **fair** comparison, we thus reimplement CODA-P using pretrained OpenAI CLIP’s ViT backbone (similar to how we compare L2P and DualPrompt in our paper). The latter reimplementation then demands memory replay to catch up with the performance of our proposed method, thus making our comparisons fair. Lastly, as shown in Table 1, our method surpasses CODA-Prompt across all datasets.
- **Our method works without replay, replay further boosts its performance:** In Table 2 of the rebuttal pdf, we have provided a comparison of our proposed method with the SOTA models for replay-free continual learning using ViT-based (i.e., CODA-Prompt) and CLIP-based (i.e., AttriCLIP) backbones. Here, leveraging the instance-conditioned and semantically diverse prompts of AttriCLIP provides an edge. Our variant leveraging AttriCLIP further improves its performance surpassing the SOTA. In the bottom two rows of the table, we ablate the role of our proposed language-aware distribution regularization and weight init. components and find that the former is crucial in avoiding forgetting in this setting.
- **Our method is agnostic of the exemplar selection strategy used for replay:** In Section 5 in the main paper, we discuss that our proposed method is agnostic of the strategy used to select exemplars for memory replay. To support this, in Table 18, Appendix D.2, we show that our method works well even with entropy-based exemplar selection strategy – a setup where deterministic methods generally lag due to their poor predictive confidences [1].

References:

[1] Chaudhry, Arslan *et al.* “Riemannian Walk for Incremental Learning: Understanding Forgetting and Intransigence.” ECCV 2018.

---

### Decision · Program_Chairs · 2024-09-25

**Decision:**

Accept (poster)

**Comment:**

Initially, the paper receives three negative scores and one positive score; after the rebuttal, one reviewer increases the score to borderline accept. I would like to thanks the authors for providing the detailed responses to the reviewers’ questions and addressed many concerns from the reviewers. The AC read through the manuscript, all reviews, and the rebuttal. Some reviewers did not actively participate in the discussion, although the AC tried several times to initiate one. The AC carefully read the authors' response and think it addresses the concerns from the reviewers well and would like to recognize the originality of the paper. The AC would like to accept this paper. The AC also strongly encourage the authors to polish the writing and incorporating the suggestions of the reviewers in the final version.